# A Yap-dependent mechanoregulatory program sustains cell migration for embryo axis assembly

Ana Sousa-Ortega [1,3], Javier Vázquez-Marín [1,3], Estefanía Sanabria-Reinoso [1], Jorge Corbacho [1], Rocío Polvillo [1], Alejandro Campoy-López [1], Lorena Buono[1], Felix Loosli [2], María Almuedo-Castillo [1] ✉ & Juan R. Martínez-Morales [1] ✉

The assembly of the embryo's primary axis is a fundamental landmark for the establishment of the vertebrate body plan. Although the morphogenetic movements directing cell convergence towards the midline have been described extensively, little is known on how gastrulating cells interpret mechanical cues. Yap proteins are well-known transcriptional mechanotransducers, yet their role in gastrulation remains elusive. Here we show that the double knockout of *yap* and its paralog *yap1b* in medaka results in an axis assembly failure, due to reduced displacement and migratory persistence in mutant cells. Accordingly, we identified genes involved in cytoskeletal organization and cell-ECM adhesion as potentially direct Yap targets. Dynamic analysis of live sensors and downstream targets reveal that Yap is acting in migratory cells, promoting cortical actin and focal adhesions recruitment. Our results indicate that Yap coordinates a mechanoregulatory program to sustain intracellular tension and maintain the directed cell migration for embryo axis development.

The assembly of the primary embryo axis is an essential event for the foundation of the body plan in all bilaterian animals. During gastrulation, the embryo axis emerges through evolutionary conserved morphogenetic movements, which operate in coordination with the lineage restriction events that lead to the specification of the three basic germ layers; ectoderm, mesoderm, and endoderm[1,2]. Basic gastrulation movements, such as emboly, epiboly, and convergence and extension, follow a similar logic in all species. In fact, the capacity to form and elongate an axis is apparently an intrinsic property of the developing cell collectives. Thus, even in the absence of stereotypic embryo cues (e.g., extraembryonic tissues and embryo geometry), isolated ESCs (Embryonic Stem Cells) have the capacity to self-organize into a rudimentary anterior-posterior (A-P) axis in vitro, as demonstrated for mammalian gastruloids[3,4] and zebrafish blastoderm aggregates[5,6].

The underlying process behind this universal capacity is the ability of precursor cells to integrate genetic and mechanical cues and migrate in a directed manner toward the embryo midline. Mechanical cues stand up among the candidate contextual inputs that may act as channeling mechanisms to maintain gastrulation as an invariant and robust process. Mechanoregulatory loops have an essential function in maintaining homeostasis during development and tissue remodeling[7]. Furthermore, dysregulated mechanical feedbacks are a common landmark in numerous pathologies, particularly in cancer[8]. In the context of gastrulation, it has been shown that mechanical strains play a conserved role in mesoderm specification both in Drosophila and zebrafish, through nuclear translocation of the transcriptional regulator ß-catenin[9]. Despite their relevance, the impact of mechanotransduction and mechanosensation on gastrulation dynamics has been scarcely explored[10–12].

[1]Centro Andaluz de Biología del Desarrollo (CSIC/UPO/JA), 41013 Sevilla, Spain. [2]Institute of Biological and Chemical Systems, Biological Information Processing (IBCS-BIP), Karlsruhe Institute of Technology, Eggenstein-Leopoldshafen, Germany. [3]These authors contributed equally: Ana Sousa-Ortega, Javier Vázquez-Marín. ✉e-mail: malmcas@upo.es; jrmarmor@upo.es

Yap proteins are well-known transcriptional regulators that are able to shuttle to the nucleus upon mechanical stimulation, being active in cells that have undergone cell spreading and inactive in round and compact cells[13]. Their ability to sense mechanical strains depends on actomyosin contractility, actin capping, and severing proteins, the integrin-talin mechanosensitive clutch, and the coupling between the extracellular matrix (ECM) and the nuclear envelope[14–16]. Initially characterized as effectors of the Hippo signaling cascade, Yap proteins play a key role both during embryogenesis, as master regulators of growth, cell specification, and survival[17]; as well as in adult organs, where they are critical for tissue repair and cancer progression[18]. More recently, an increasing number of reports have linked Yap proteins to cell rearrangements and tissue morphogenesis[19,20]. This is not surprising, given Yap/Taz transcriptional ability to modulate cytoskeletal and ECM components, as reported in mammalian cell lines[21,22]. However, despite these observations, the mechanistic link between Yap proteins and gastrulation morphogenetic movements has remained elusive.

Here we show that the simultaneous inactivation of *yap1* and *yap1b*, the two members of the Yap family in medaka[23], results in a complete failure to assemble the posterior half of the embryo axis. The analysis of the transcriptional program activated by Yap proteins at gastrulation stages indicates that the general specification of the germ layers does not depend on Yap function. In contrast, Yap proteins activate the expression of genes encoding for cytoskeletal regulators, ECM, and focal adhesion components; suggesting a direct role in controlling the morphogenetic behavior of the gastrulating precursors. Quantitative live-imaging analysis of cell displacement trajectories confirmed that Yap proteins are required for dorsally directed cell migration towards the midline. By following Yap activity using an in vivo Tead sensor (*4xGTIIC:GFP*) and the expression of Yap downstream targets, we show that Yap is active in dorsally migrating precursors, rather than in compacted cells at the developing axis. In the absence of Yap function, mutant cells show reduced focal contacts and

cortical actin recruitment, and fail to acquire the characteristic flattened morphology of the wild-type (WT) migratory cells. We also show that in the context of gastrulating cells Yap activation depends on actomyosin contractility. These observations point to the existence of a Yap-dependent mechanoregulatory feedback that ensures the efficient convergence of the precursors to the midline; a mechanism likely conserved in many other homeostatic and developmental processes.

## Results

### Yap paralogs are required for proper axis development in medaka

Gastrulation relies on extensive cellular rearrangements responsible to place the three germ layers in their correct topological position, while directing the formation of the embryo body axis[24]. Previous work in different vertebrate species, including our work in medaka[23], hinted to a potential role for the mechanotransducer Yap during axis development. To gain insight into the role of Yap family proteins in this process, we focused on the phenotypic consequences of mutating both *yap* paralogs, *yap1* and *yap1b* in gastrulating medaka embryos. When we examined *yap1−/−* medaka embryos at stage 20 (hereinafter referred as 'single mutants'), we observed that somite formation was affected, and the anterior-posterior (A-P) axis was wider and shorter (Fig. 1A, B). Despite these defects, the somitogenesis recovered, and the primary embryo axis (although still shorter and wider) was eventually formed in single mutant embryos at later stages (stage 24), as previously reported[19,23] (Fig S1A–D). Remarkably, *yap1−/−;yap1b−/−* double mutants (hereinafter *yap* double mutants) displayed much stronger developmental defects, as posterior axis assembly and somites were not apparent at stage 20 (Fig. 1A, C). Moreover, *yap* double mutants did not survive after stage 23. To further characterize the mutants phenotype, we performed a DAPI and Phalloidin staining to visualize nuclei and filamentous actin during late gastrulation (stages 16-17) (Fig. 1D–F). Confocal analysis showed how, in WT embryos, A-P axis assembly becomes apparent and actin network

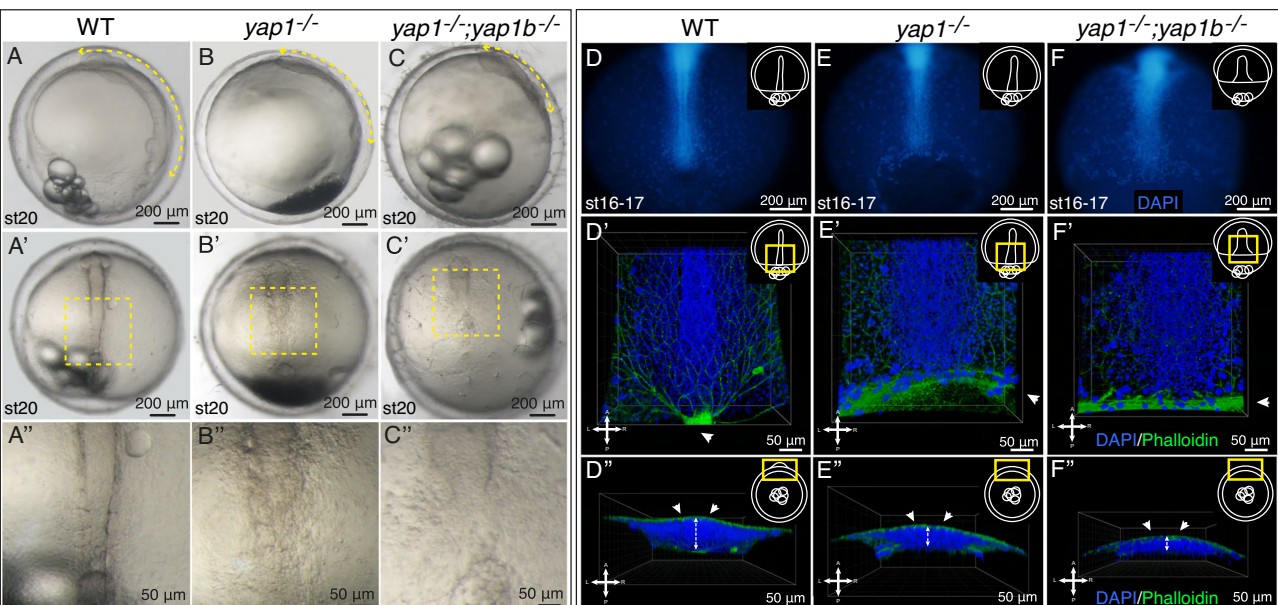

**Fig. 1 | Analysis of axis assembly in *yap* mutants. A–C** Brightfield images of WT, *yap1−/−* and *yap1−/−;yap1b−/−* embryos at stage 20 (post gastrulation stage) **A–C**. Yellow double-headed arrows highlight the shortening and the widening of the A-P axis **A–C**. **A″–C″** correspond to the magnifications indicated with a yellow square in their corresponding **A′–C′** images. **D–F** DAPI staining, in which nuclei are labeled in blue, was performed in WT, *yap1−/−* and *yap1−/−;yap1b−/−* embryos fixed at stage 16–17 (late gastrula) (**D–F**). Whole embryos are shown under the fluorescent stereo microscope. DAPI and Phalloidin immunostained confocal images show the posterior axis of WT, *yap1−/−* and *yap1−/−;yap1b−/−* embryos at stage 16, 17 (nuclei in blue and filamentous actin in green) (**D′**, **E′**, and **F′**). XZ projections of the DAPI and Phalloidin immunostaining, showing the D–V height of the axis of WT, *yap1−/−* and *yap1−/−;yap1b−/−* embryos (**D″**, **E″** and **F″**) Embryos' orientations are indicated with a cross (A: anterior, L: left, R: right, P: posterior). Yellow rectangles in schematic embryo representations indicate the area depicted in each image. Scales bars are 200 μm (**A–C**, **A′–C′** and **D–F**) and 50 μm (**A″–C″**, **D′–F′** and **D″–F″**).

concentrates at the epiboly front, particularly at the closing blastopore (Fig. 1D–D'', S1E and Supplementary Movie 1). In contrast, in *yap* single mutants, actin staining appeared more diluted at the delayed blastopore margin, as previously reported[19,23], and a decreased density of cells at the midline was observed (Fig. 1E–E'', S1E and Supplementary Movie 1). In agreement with our previous findings, *yap* double mutants completely fail to assemble their posterior part, displaying a significantly reduced dorso-ventral (D-V) accumulation of cells at the midline, and without an apparent formation of the presumptive neural plate and paraxial mesoderm masses (Fig. 1F–F'', Fig S1E and Supplementary Movie 1).

Yap has a crucial role in proliferation and cell survival[18,25,26], and an increase in cell death in *yap* mutants has been reported after neurulation at stage 20–22[19,23]. Therefore, we evaluated if a possible explanation for the observed phenotype was a change in cell death and proliferation rates in *yap* double mutants during gastrulation. To this end, we quantified the number of apoptotic and proliferative cells labeled by caspase-3 and pH3, respectively, in WT and *yap* double mutants in late gastrulation embryos at stage 16. We did not observe significant differences in cell death nor proliferation density between WT and *yap* double mutant embryos at these earlier stages of development (Fig S1F-I). Given that variations in cell death and proliferation could be ruled out as responsible for the gastrulation defects, we decided to explore alternative mechanisms behind the axis assembly failure.

## Yap is needed for a correct cell migration during gastrulation

In teleost embryos, axis assembly is achieved by dorsal migration and lateral intercalation of the precursors at the midline[1], a process that is largely conserved in medaka[27]. Since we observed a clear failure in midline cell stacking in our *yap* mutants, we asked ourselves if directed cell migration was altered. To assess that, we analyzed cell trajectories during gastrulation using live-imaging in medaka embryos. In WT embryos, cells move dorsally from the lateral regions towards the central axis in a straight manner (Fig. 2A and Supplementary Movie 2). *Yap* single mutants seem to display lower accuracy in their directionality and cells are slightly delayed when reaching the embryo axis (Fig. 2B and Supplementary Movie 2). Displacement defects are markedly accentuated in *yap* double mutant embryos, where many cells display abnormal trajectories and deficient migration towards the midline (Fig. 2C and Supplementary Movie 2). To further confirm these observations, we performed a high throughput cell-tracking analysis to measure the main parameters involved in directed migration. First, we measured cell displacement, which quantifies how much a cell moves from its start point, and represented this parameter with a color gradient (Fig. 2D, E). We could observe that unlike in WT embryos, long-displacing cells (i.e., red and yellow trajectories) were rarely detected in *yap* double mutants, whereas short-displacing cells (i.e., blue and green trajectories) predominate (Fig. 2D, E). The statistical analysis of these measurements confirmed our observations, as displacement mean values were significantly lower in *yap* double mutants compared to WT embryos (Fig. 2F). The second parameter we evaluated was the migratory persistence of cells, which quantifies how long a cell keeps the same direction. This analysis revealed that the migratory persistence of *yap* mutant cells is significantly reduced compared to WT cells (Fig. 2G). Finally, we also measured the cell trajectory length and the mean velocity. Similarly to the previous parameters, we could clearly observe that WT cells move faster and through longer tracks than *yap* mutant cells (Fig. 2H, I). However, when we analyzed convergent and extension movements of cells at the midline, these migratory parameters were not significantly altered. Only the displacement of *yap* mutant cells is slightly reduced, which might be explained by the fact that the mutants have a shorter embryo axis (Fig S2). Taken together, these results indicate that Yap proteins play an essential role to direct cell migration in gastrulating embryos, specifically of dorsal cells converging to the midline.

## Yap transcriptional program primarily regulate cytoskeleton organization and cell adhesion components

Diverse molecular cues have been shown to direct polarized cell movements during gastrulation, such as cell-to-cell adhesion, interaction with the ECM, or chemotaxis[1]. On the other hand, Yap proteins have been shown to activate context-dependent transcriptional programs[28,29]. Therefore, in order to get a complete picture on how Yap might be directing cell trajectories, we performed a comparative RNA-seq analysis of WT, *yap* single and double mutant embryos at mid-late gastrulae stage (stage 16) (Fig. 3, S3). Using this approach, we identified 717 and 1178 differentially expressed genes (DEGs) in *yap* single and double mutants compared to WT embryos, respectively (Fig. 3A, S3A, Supplementary Data 1). Principal components analysis (PCA) of the obtained results showed differential clustering of WT samples vs *yap* single and double mutants (Fig S3B), supporting our previous finding that both *yap1* and *yap1b* control very similar transcriptional programs[23]. For that reason, we focused further analyses on *yap* double mutants most severe phenotype.

To understand the mechanisms underlying Yap activity, we next studied Gene Ontology (GO) terms enrichment in the DEGs in *yap* double mutants *vs* WT (Fig. 3B, C and S3C, D; Supplementary Data 2). We explored four different GO categories: molecular function, cellular component, biological process, and KEGG pathway. In the molecular function category, we identified integrin binding as the most significantly enriched GO term, followed by others such as acting binding or cell adhesion (Fig. 3B). Very consistent results were also obtained for the remaining GO categories (Fig. 3C and Fig. S3C, D). Thus, significantly enriched GOs terms identified in *yap* double mutants were related to cell-ECM adhesion (i.e., focal adhesion (FA) or collagen-containing ECM), Hippo signaling, and actin cytoskeleton regulation/organization (Fig. 3B, C, Fig S3C, D; Supplementary Data 2). Similar GO enrichments were obtained when DEGs between yap single mutants *vs* WT were considered (Fig S3E–H). These data indicate that Yap paralogs primarily regulate the expression of actomyosin cytoskeleton and ECM-cell adhesion components (Supplementary Table 1; Supplementary Data 3).

Cellular rearrangements during gastrulation are often coordinated with lineage restriction and germ layers specification. To verify whether cell fate is compromised in *yap* mutants, we examined in more detail our RNA-seq datasets, which in principle did not yield significantly enriched GO terms consistent with that hypothesis (Fig. 3B, C, Fig S3C–H). For further confirmation, we checked the expression of a battery of 10 conserved specifiers of each germ layer. No significant differences were observed for key mesoderm, endoderm, and neuroectoderm markers when their expression was compared between *yap* double mutant and WT embryos (Fig. 3D). The only genes that appeared significantly downregulated correspond to early non-neural ectoderm (i.e., epidermal) specifiers (Fig. 3D). To confirm these observations, we compared the expression patterns of a mesoendodermal marker, *no-tail* (*ntl*), an endodermal marker, *goosecoid* (*gsc*), and an ectodermal marker, *sox3* (Fig. 3E). In agreement with our RNA-seq data, these three germ-layer markers did not appear downregulated in *yap* mutants. However, their expression patterns appeared wider in *yap* mutants, which suggests a failure in cell convergence, in line with the defective migration of cells that we observed (Fig. 3E). Taken together, these analyses indicate that what is behind the cell migration defects is a failure in the activation of the genetic program controlling cytoskeleton reorganization and cell adhesion, rather than a general problem in cell fate acquisition.

To gain further insight into the genetic program controlled by Yap proteins, we compared our RNA-seq data with DamID-seq results we previously obtained in stage 16 medaka embryos[23]. Using the DamID-seq technique, we generated maps of chromatin occupancy for Yap1 and Yap1b in gastrulating embryos. Then, by cross-comparing genes neighboring Yap paralogs binding sites with our list of DEGs, we could determine which of these genes are potential direct targets (i.e., genes

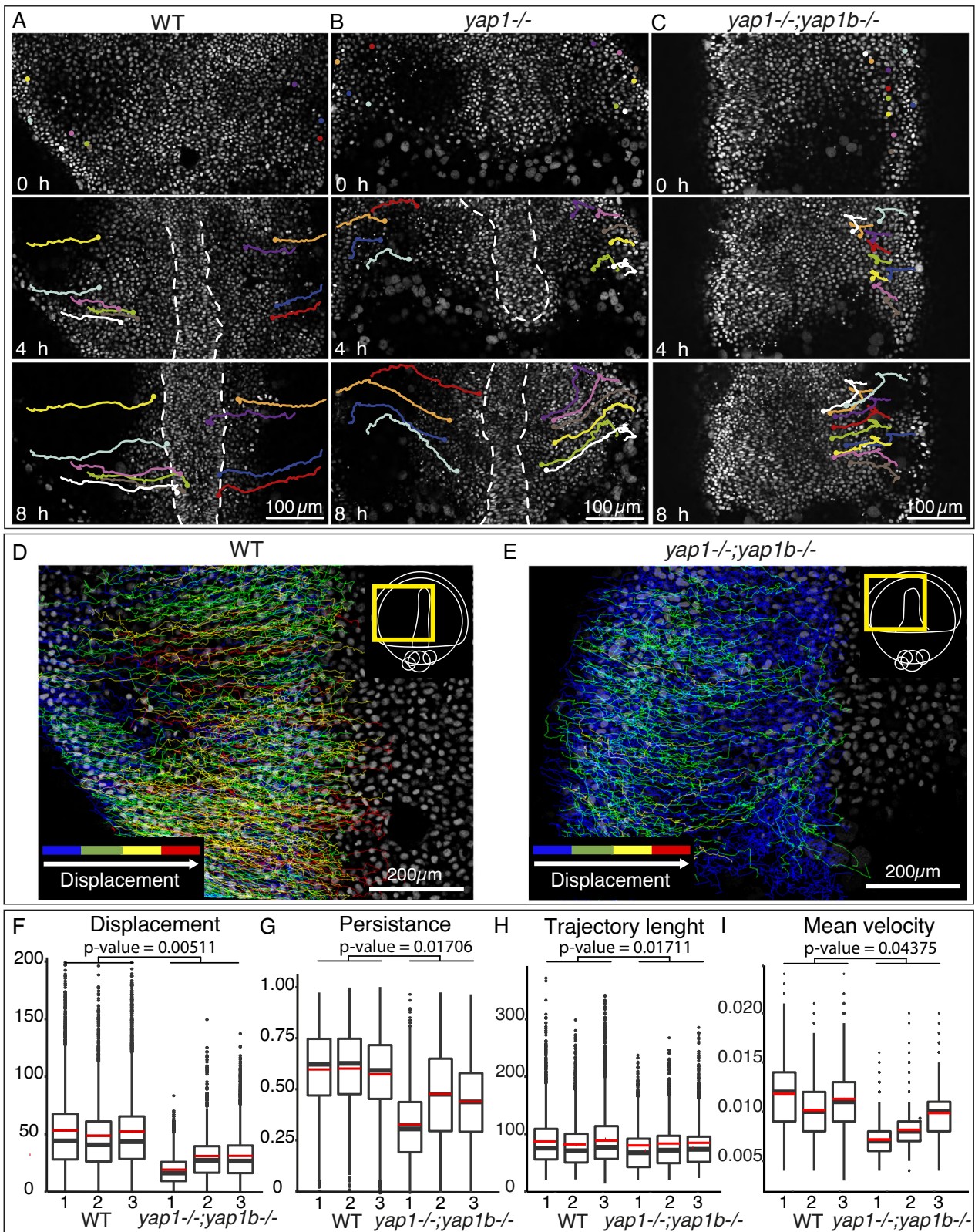

downregulated in *yap* double mutants in which Yap binds to nearby regulatory regions according to our DamID-seq results). We observed that a significantly high percentage of these DEGs are potentially direct targets of Yap1/Yap1b (Fig. 3F), and that these genes have a very similar list of associated enriched GOs terms as the entire set of DEGs (Supplementary Data 2). Importantly, we could confirm within them relevant regulators of the cytoskeleton, such as *marcksl1b*; structural ECM

encoding genes, such as *lamc1*; well-known Yap targets such as *ccn1/ cyr61*; and Yap regulators such as *src* among the targeted genes (Fig S4 and Supplementary Data 2).

**Yap is active in migratory cells converging to the midline**
Cell migration depends on actin polymerization at the leading edge to drive protrusions that adhere to the substrate through FAs[30].

**Fig. 2 | Defective directed cell migration in gastrulating *yap* mutant embryos.** **A**−**C** Manual tracking of representative cells of gastrulating WT (**A**), *yap1⁻/⁻* mutant (**B**), and *yap1⁻/⁻;yap1b⁻/⁻* mutant embryos (**C**) injected with *Histone2B*::GFP for nuclei visualization. Still images from supplementary movie 2 at 0 h, 4 h, and 8 h are shown. Each cell trajectory is represented with a color line. When present, the midline is represented with white, dashed lines. **D**−**E** Total individual cell migratory tracks over 8 h in WT and *yap1⁻/⁻;yap1b⁻/⁻* embryos. The color code of the trajectory lines indicates the cells' displacement (distance between the start and end position of a cell). Displacement values were represented as: Blue = 20–50; Green = 50–80; Yellow = 80–130; Red>130. Yellow rectangles in schematic embryo representations indicate the area depicted in each image. **F** Quantification of cell displacement in WT and *yap1⁻/⁻;yap1b⁻/⁻* embryos. *P* value = 0.005111. **G** Quantification of cell migratory persistence, measuring for how long a cell keeps the same direction of movement, in WT and *yap1⁻/⁻;yap1b⁻/⁻* embryos. *P* value = 0.01706. **H** Quantification of cell trajectory length, measuring the total length of a cell trajectory *P* value = 0.01711. **I** Quantification of the cell mean velocity, measuring the distance between two cells' positions divided by the time difference, in WT and *yap1⁻/⁻;yap1b⁻/⁻* embryos. *P* value = 0.04375. Boxes represent the quartiles; the whiskers indicate the maximum and minimum values. Red and black lines indicate the median and the mean, respectively. To analyze whether experimental groups were significantly different, a variance test followed by a two-sided Student's *t* tests were performed on the means of WT and *yap1⁻/⁻;yap1b⁻/⁻* embryos (*n* = 3 embryos; *n* = 2562 cells were examined in WT1, *n* = 2433 cells were examined in WT2, *n* = 3978 cells were examined in WT3, *n* = 2050 cells were examined in *yap1⁻/⁻;yap1b⁻/⁻* mutant 1, *n* = 1391 cells were examined in *yap1⁻/⁻;yap1b⁻/⁻* mutant 2, *n* = 2591 cells were examined in *yap1⁻/⁻;yap1b⁻/⁻* mutant 3). Scale bars are 100 μm (**A**−**C**) and 200 μm (**D**−**E**). Source data are provided as a Source Data file.

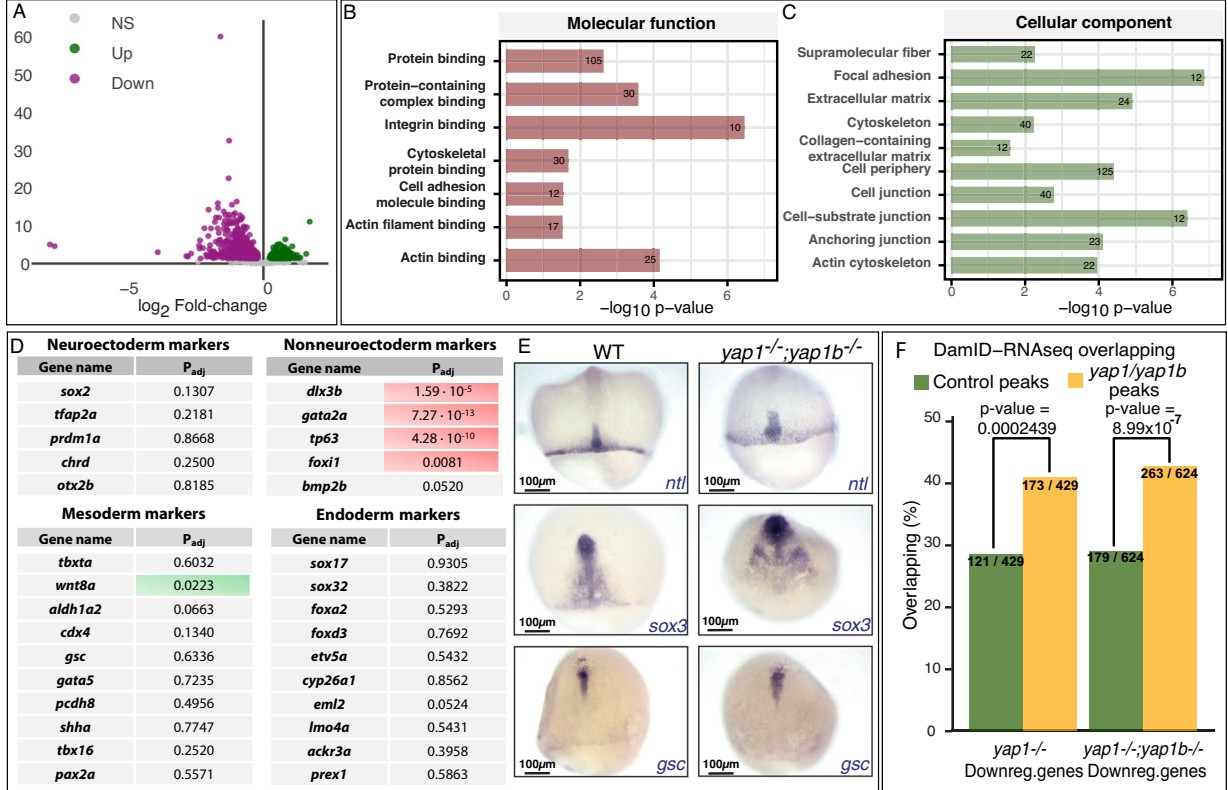

**Fig. 3 | Characterization of Yap-dependent transcriptional program. A** Volcano plot graph showing differentially expressed genes (DEGs) between WT to *yap1⁻/⁻;yap1b⁻/⁻* embryos. Gray dots: no differentially expressed genes; Green dots: upregulated genes in *yap1⁻/⁻;yap1b⁻/⁻*; Magenta dots: downregulated genes in *yap1⁻/⁻;yap1b⁻/⁻*. Differential gene expression analysis was carried out using the R package DESeq2, using by default Wald test and Benjamini−Hochberg correction (*p*adj <0.05; −log FC = 1). **B**, **C** Gene Ontology (GO) enrichment of the DEGs in *yap1⁻/⁻;yap1b⁻/⁻* embryos compared with WT, classified in molecular function (**B**) and cellular component (**C**). *n* = 3 embryos. gProfiler was used for this analysis (see Methods section). **D** Differential expression of the 10 most conserved markers of each germ layer in *yap⁻/⁻;yap1b⁻/⁻* compared to WT embryos. Adjusted *p* values are shown. Red indicates significantly down-expressed genes in *yap1⁻/⁻;yap1b⁻/⁻* embryos compared with WT; Green indicates significantly upexpressed genes in *yap1⁻/⁻;yap1b⁻/⁻* embryos compared with WT. **E** *ISH* analysis of the expression of *ntl* (mesodermal marker), *sox3* (ectodermal marker), and *gsc* (endodermal marker) in WT and *yap1⁻/⁻* and *yap1⁻/⁻* embryos at stg 16. Scale bars = 100 μm. **F** Quantification of the overlap between genes associated with regions targeted by Yap1/Yap1b, as determined by DamID-seq (Vazquez-Marin et al., 2019), and DEGs between WT and *yap1⁻/⁻* (*P* value = 0.0002439) or WT and *yap1⁻/⁻;yap1b⁻/⁻* (*P* value = 8.99 × 10⁻⁷) embryos. *n* = 3 embryos. The statistical significance of the differences between control and Yap1/Yap1b-targeted regions was calculated applying a two-sided *Z* test. Source data are provided as a Source Data file.

Our observation that Yap activates the transcriptional programs controlling cytoskeleton and FA components suggests that its activity may be required in migratory cells. To confirm this point, we checked the spatiotemporal dynamics of Yap activation by following both the transcriptional activity of a Tead/Yap sensor, and the expression of *marcksl1b*, a bona-fide Yap target identified in our DamID-seq and RNA-seq analyses[23]. Tead co-activators interact with Yap proteins acting as main mediators of their transcriptional response[23,28,29,31]. To monitor Yap activation, we generated a new medaka transgenic line in which the Tead-responsive *4xGTIIC* enhancer, previously tested in zebrafish, was coupled to GFP (*4xGTIIC::GFP*)[32,33]. To validate this transgenic line, we first checked that the GFP signal was undetectable in *yap* mutants (Fig S5A). Then, by injecting *yap1* mRNA fused to the mCherry reporter gene (*yap1::mCherry*), we confirmed that the cells with higher levels of nuclear Yap:mCherry are displaying higher intensity for the Tead/Yap reporter signal (Fig S5B). These results corroborate that the *4xGTIIC::GFP* transgenic line responds specifically to Yap signaling. Using this tool, we could then follow in vivo the dynamics of Yap

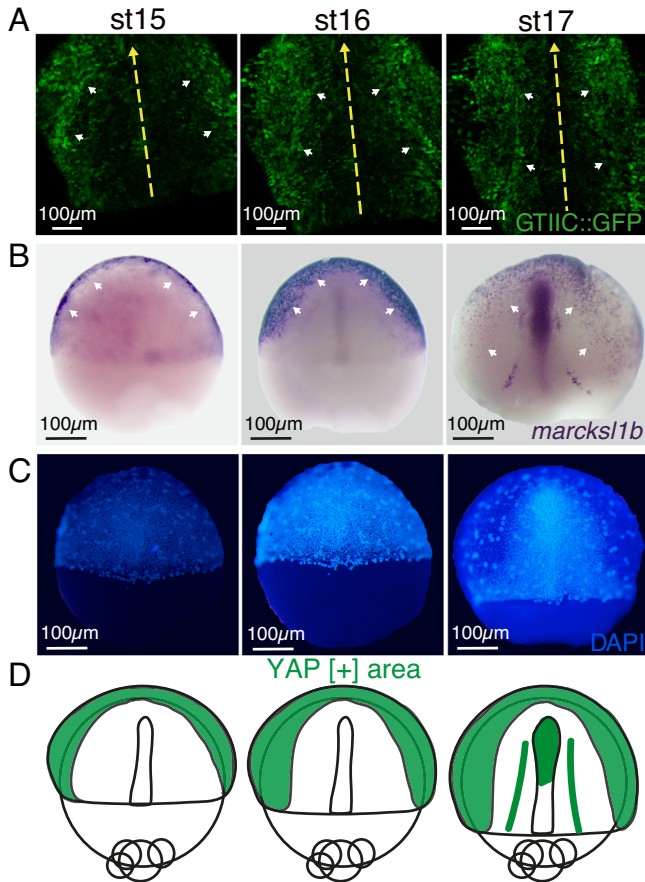

**Fig. 4 | Yap activation in migratory precursors. A** Still images from the confocal time-lapse analysis (supplementary movie 3) of transgenic embryos for the Tead/Yap sensor *GTIIC::GFP* at stages 15, 16, and 17. White arrowheads point to the GFP-positive cells. Yellow arrows indicate A-P axis (**B**) ISH analysis of the expression of *marcksl1b* in WT embryos at 15, 16, and 17. White arrowheads point to the cells expressing *marcksl1b*. **C** DAPI staining of the embryos in **B**. **D** Schematic representation of the expression of *marcksl1b* (Yap activation area) in green. Scale bars are 100 μm (**A–C**).

activation during gastrulation by live confocal imaging (Supplementary Movie 3). Interestingly, we observed that Yap is active in cells migrating towards the midline, rather than in the midline itself, where cell density is higher (Supplementary Movie 3 and Fig. 4A). To corroborate these observations, we also examined the expression pattern of *marcksl1b* by in situ hybridization. Marcksl1b is a protein involved in cell motility, as it regulates actin cytoskeleton dynamics as well as filopodium and lamellipodium formation[34]. Matching our observations with the Tead reporter line, we saw that in gastrulating embryos, *marcksl1b* is expressed mainly in lateral cells (Fig. 4B). As expected, we could confirm that *marcksl1b* expression is largely decreased in *yap1* mutant embryos (Fig S5C). These results indicate that during gastrulation, Yap is specifically active in cells that are moving towards the midline, while inactive in the more compact cells forming the embryo axis (Fig. 4C, D). Thus, we concluded that the migration defects observed specifically in dorsally-convergent *yap* mutant cells are consistent with the absence of Yap activation in these migratory cells.

In agreement with our previous findings[23], we confirmed that *yap1* is ubiquitously expressed at early gastrula (stage 15), and that as gastrulation progresses its expression is enriched at the condensed axis (Fig S5D). This discrepancy between *yap* expression and Yap activation indicates that Yap signaling inhibition depends on post-transcriptional regulation. It has been described that Yap activity can be directly inhibited by cell density[35,36]. To assess whether an anti-correlation

between Yap activation and cell compaction was also significant in the gastrulation context, we examined density maps in relation to *marcksl1b* expression (Fig. 5A–C). We could determine that the distance between neighbors is significantly higher (lower cell density) in Yap-active areas than in Yap-inactive areas (Fig. 5D), suggesting that cell density may act as a modulator of Yap activity also during gastrulation. To further explore this hypothesis, we developed a method to increase cell density by extracting yolk material from 50% Epiboly WT embryos using microcapillaries. This approach resulted in an averaged reduction of 40% of the embryo perimeter that was accompanied by a significant increase in the inner mass cell density (~30 cells more per 100 μm²) (Fig. 5E, I). Interestingly, this increase in cell density observed in the yolk-reduced embryos, correlated with a decrease in Yap activity, evidenced by a lower *marcksl1b* expression and Tead/Yap sensor activation (Fig. 5F, J, K). Altogether, these findings suggest that cell density acts as a negative regulator of Yap activity in midline regions of gastrulating embryos.

## Yap promotes cortical actin recruitment and focal adhesions assembly in migratory cells

We showed that Yap proteins are active in dorsally converging cells, in which they modulate the expression of cytoskeletal and ECM adhesion components. To investigate the recruitment of these components to the cell cortex in convergent cells, embryos were injected at the one-cell stage either with *Utrophin:GFP* mRNA, to label filamentous actin, or *Pax::mKate* mRNA, to reveal FAs assembly. Then, the distribution of these tracers was examined using high-resolution microscopy in WT and *yap* double mutants, focusing our attention on dorsal migratory cells of the inner mass (Fig. 6), in which Yap is active (Fig. 4). These inner mass cells display a monolayer distribution, positioned between the large polygonal cells of the enveloping layer (EVL) and the yolk syncytial layer (YSL). WT inner mass cells presented strong accumulation of filamentous actin (Fig. 6A) and showed the spreading shape typical of migratory cells, with extended plasma membrane and protruding filopodia and lamellipodia. On the contrary, *yap* mutant cells accumulated less cortical actin, displaying a rounded shape with less noticeable protrusions (Fig. 6A, C, D). We also observed that WT inner mass cells display FA stripes and foci, which are very reduced in *yap* double mutant cells (Fig. 6A). This result suggests that Yap is essential for FA maturation, as these structures enlarge as they mature[37]. By looking at the XZ projections of *Paxillin:mKate* in WT cells, we observed that FA clusters tend to accumulate at the surface contacting the YSL (Fig. 6A'), thus suggesting that inner mass cells are migrating preferentially over the yolk surface. *Yap* mutant cells clearly lack these polarized adhesion clusters, which are significantly shorter (Fig. 6A', E; S6C).

To get further insight into the cells' morphology, we performed a similar experiment but now injecting *Utrophin:GFP* together with *Lyn-tdTomato (LynTm)* mRNA, to visualize the plasma membrane. These results confirmed the cortical accumulation of filamentous actin in WT cells, which presented multiple filopodia and membrane ruffles (Fig. 6B–D). In contrast, *yap* mutant cells displayed a rounded morphology with lower number of filopodia (Fig. 6B–D). By examining XZ projections of the nuclei, we observed that WT nuclei tend to appear less spherical than *yap* mutant cells (Fig. 6B', S6A, Supplementary Movie 4). Since mechanical coupling to the ECM and nuclear deformation are required for Yap nuclear shuttling[15,16], we decided to explore this observation further. To this end, the 3D nuclei morphology of WT and *yap* mutant cells was measured, considering two main morphological parameters, flatness, and sphericity. We found that *yap* mutant nuclei display a lower flatness index and are significantly more spherical than those of WT cells (Fig. 6F, S6D,E). From these measurements, we hypothesized that the noticeable reduction of cortical filamentous actin and FAs observed in *yap* mutant cells may lead to a decrease in intracellular tension, which is reflected in a more relaxed

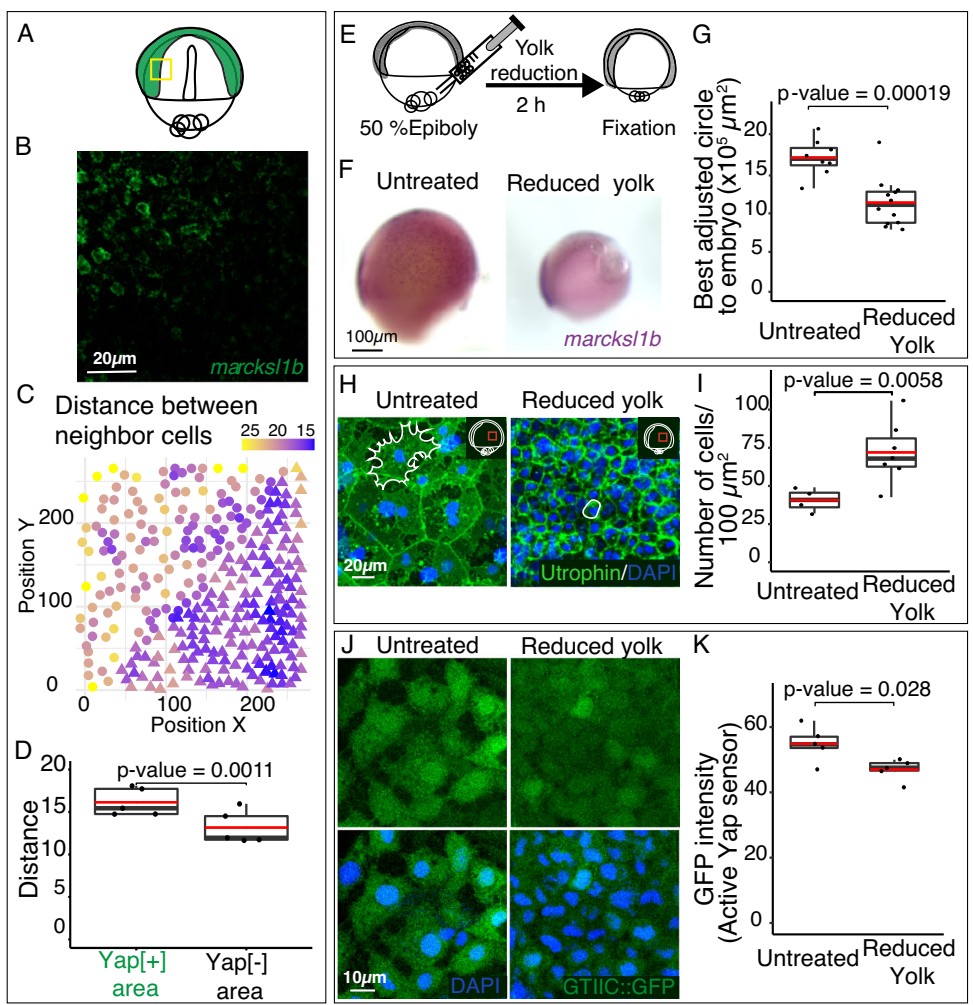

**Fig. 5 | Yap activation is inhibited by cell density. A** Schematic representation of Yap activation in green in a stage 16 embryo. **B**, **C** Confocal image of *marcksl1b* fluorescent ISH at the Yap activation margin in stage 16 embryos (**B**) and its corresponding cell density analysis (**C**). The position of the region considered in **B** is indicated with a yellow rectangle in the scheme in **A**. The XY position of the nuclei's centroids were represented. Circle: nuclei localized in a *marcksl1b* positive area; triangle nuclei localized in a *marcksl1b* negative area. The color gradient refers to the mean of the distance between a cell and its five closest nuclei **C**.
**D** Quantification of the mean distance between nuclei in the *marcksl1b*-positive area (Yap active) and in the *marcksl1b*-negative area (Yap inactive). *P* value = 0.0011. *n* = 5 embryos. **E** Schematic representation of the yolk removal protocol.
**F** ISH analysis of the expression of *marcksl1b* in control and reduced yolk embryos.
**G** Quantification of embryo size, as determined by the area of the best-adjusted circle to control and yolk-reduced embryos. *P* value: 0.00019. *n* = 8 control embryos, *n* = 12 reduced yolk embryos. **H** Confocal microscopy images of dorsally converging cells from control and yolk-reduced embryos injected with *Utrophin::GFP* and stained with DAPI. A schematic representation of the embryo indicating the area of interest (red rectangle) is shown in the bottom left side of the image. Cell shapes are represented with white lines. **I** Quantification of cell density in control and yolk-reduced embryos. *P* value: 0.0058. *n* = 4 control embryos, *n* = 7 reduced yolk embryos. **J** Confocal microscopy images of dorsally converging cells stained with DAPI from control and reduced yolk transgenic embryos for the Tead/Yap sensor *GTIIC::GFP*. **K** Quantification of GFP signal intensity in control and reduced yolk transgenic embryos for the Tead/Yap sensor *GTIIC::GFP*. *P* value: 0.028. *n* = 5 control embryos, *n* = 5 reduced yolk embryos. Boxes represent the quartiles; the whiskers indicate the maximum and minimum values. Red and black lines indicate the median and the mean, respectively. Points indicate independent embryos. Two-sided Student's *t* tests were performed to evaluate statistical significances. Scales bars are 20 μm (**B**), 100 μm (**F**), 20 μm (**H**), and 10 μm (**J**). Source data are provided as a Source Data file.

and rounded nuclei and cell morphology. To examine if these morphological defects depend on cell-ECM adhesion, we tested if the overexpression of *paxillin*, a central mechano-responsive FA component[38], was sufficient to alleviate *yap* mutants cellular phenotype. Injection of *paxillin::mKate* mRNA (300 pg per embryo) at one-cell stage in double mutant embryos partially rescued the defects at a cellular level; including spread cell morphology, cell compactness, filopodia number, and nuclear morphology (Fig S7). These observations further suggest a FA-dependent reduction of intracellular tension in our mutants, which is in agreement with previous observations indicating reduced tissue tension in medaka mutant embryos for *yap1*, as well as in *yap/taz* knockdown embryos in zebrafish[19].

All these data suggest that Yap activity is promoting the formation and maturation of FAs and the polymerization of cortical actin in migratory cells. Thus, in the absence of Yap activity, cells would be unable to establish mature FAs and actin bundles, which are essential to respond to ECM cues.

## Yap senses and activates intracellular tension suggesting a positive feedback loop

Our results indicated that Yap activity regulates FAs and actomyosin cytoskeleton, which allows coupling intracellular tension to the ECM. In turn, Yap transcriptional regulators have been characterized as mechanosensors/mechanotransducers[13,14]. Therefore, we wondered if Yap transcriptional response depends on mechanical strains, thus closing a mechanoregulatory feedback loop in gastrulating precursors. Integrins transmit information on the rigidity of the ECM through the Rho/Rock pathway, which modifies the F-actin cytoskeleton and

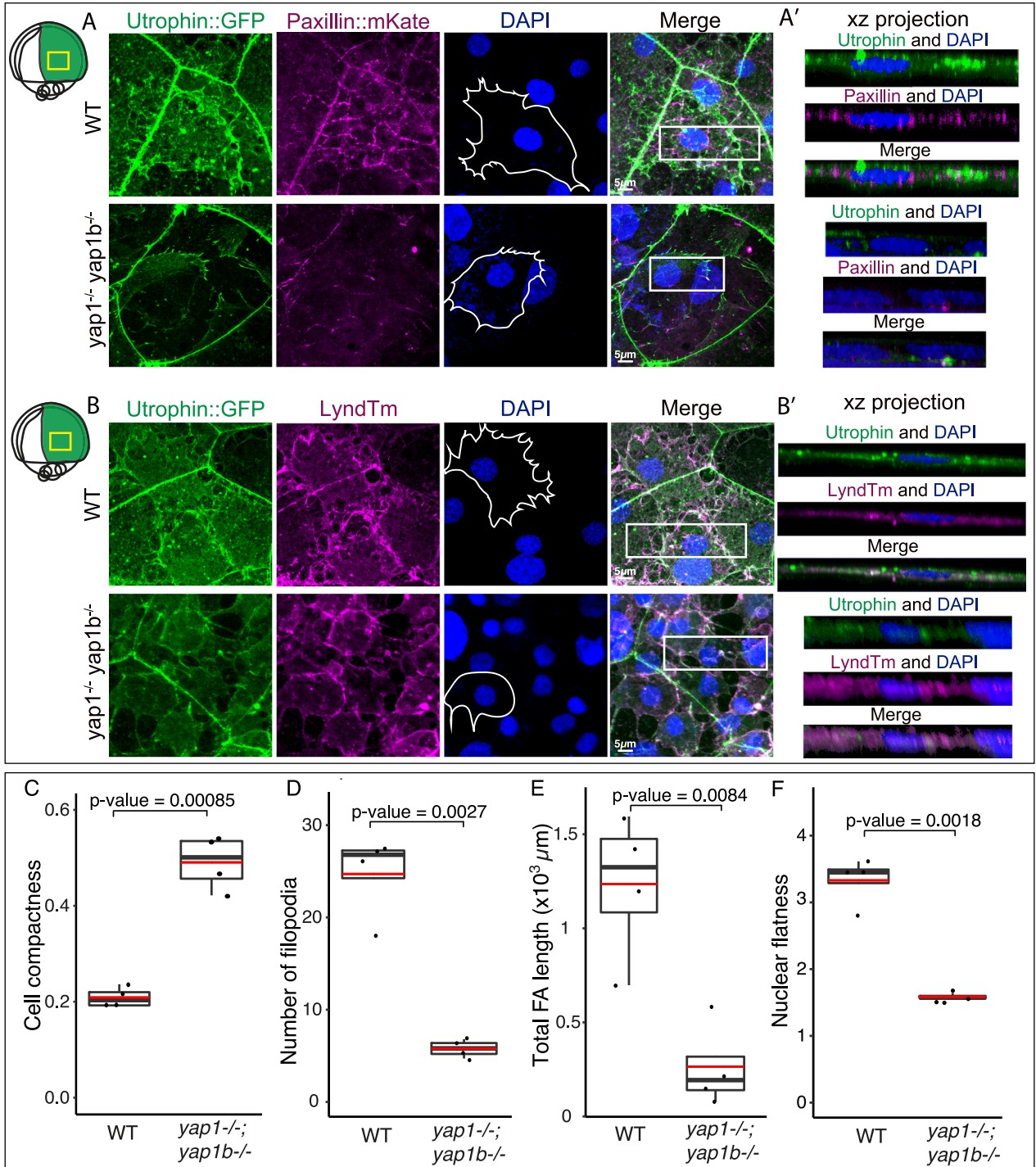

**Fig. 6 | Defective recruitment of cortical actin and focal adhesions in *yap* mutants. A, B** Confocal microscopy images of dorsally converging cells from WT and *yap1⁻ᐟ⁻;yap1b⁻ᐟ⁻* embryos injected with *Utrophin::GFP, Paxillin::mKate* (**A**), *LyndTm* (**B**) and stained with DAPI. XZ projections from the sections indicated with white rectangles are shown (**A'**, **B'**). Schematic representation of the embryo indicating the area of interest with a yellow rectangle is shown on the upper left side of the panel. Cell shapes are represented with white lines in the images corresponding to DAPI. **C** Quantification of average cell compactness, as determined by the ratio between the cell area and the area of the circle having the same perimeter, in WT and *yap1⁻ᐟ⁻;yap1b⁻ᐟ⁻* embryos. *P* value: 0.00085.
**D** Quantification of the average number of filopodia in WT and *yap1⁻ᐟ⁻;yap1b⁻ᐟ⁻*

embryos. *P* value: 0.0027. **E** Quantification of the total focal adhesion length, as determined by the sum of all focal adhesions length within a 135 × 135 µm² region of lateral converging cells, in WT and *yap1⁻ᐟ⁻;yap1b⁻ᐟ⁻* embryos. *P* value: 0.0084.
**F** Quantification of average nuclei flatness, which refers to the ratio between the second and the third axis of an ellipsoid, in WT and *yap1⁻ᐟ⁻;yap1b⁻ᐟ⁻* embryos. *P* value: 0.0018. See Figure S6 for a detailed plot. Boxes represent the quartiles; the whiskers indicate the maximum and minimum values. Red and black lines indicate the median and the mean, respectively. Points indicate independent embryos. *n* = 4 embryos. Two-sided Student's *t* tests were performed to evaluate statistical significance. Scale bars = 5 µm. Source data are provided as a Source Data file.

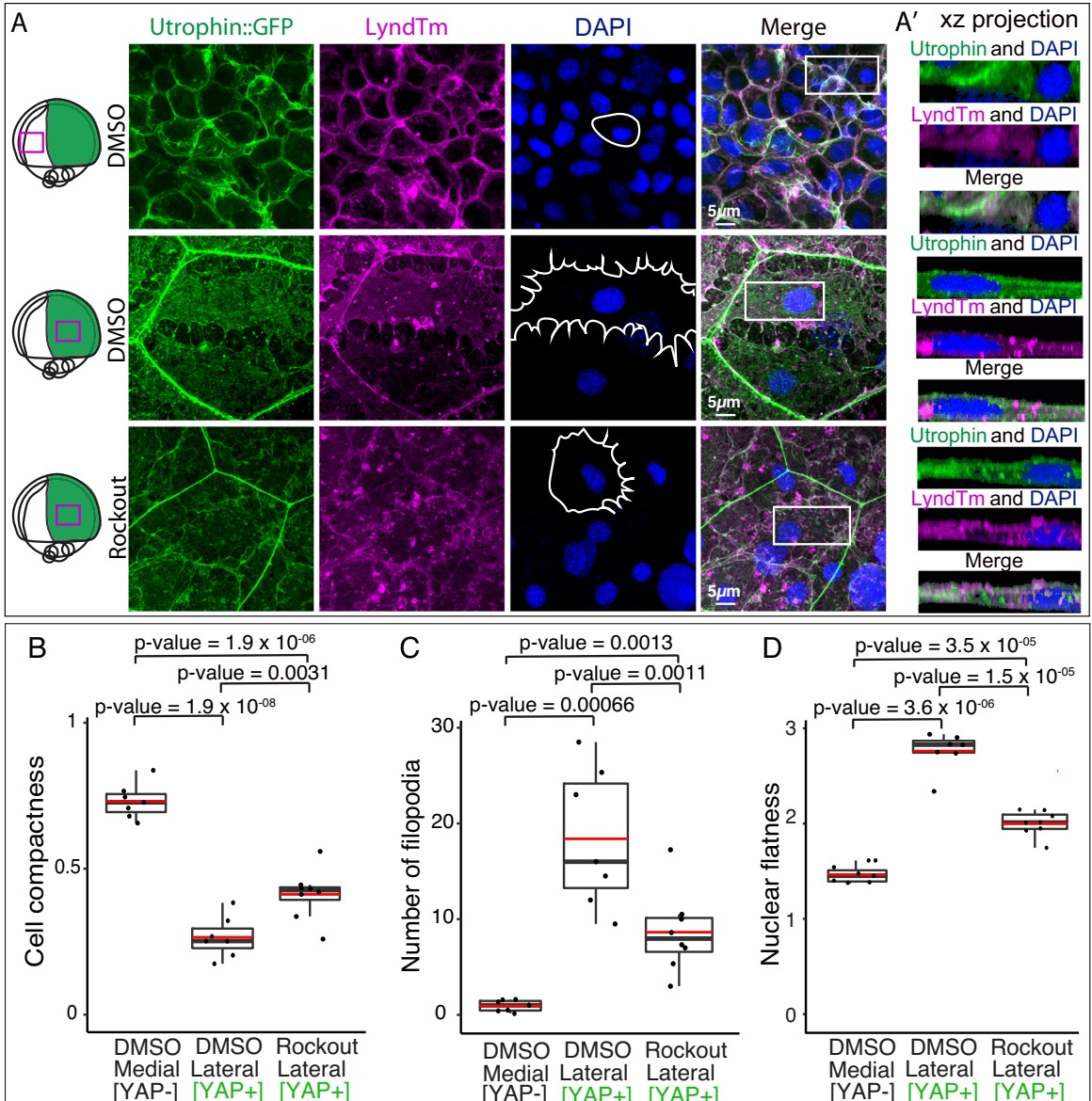

**Fig. 7 | Cell morphology and cortical actin recruitment depend on Yap activity.** **A** Confocal microscopy images of medial and lateral (DMSO) and lateral (Rockout-treated) converging cells from embryos injected with *Utrophin::GFP*, and *LyndTm* (**B**) and stained with DAPI. XZ projections from the sections indicated with white rectangles are shown (**A'**). Schematic representation of the embryo indicating the area of interest with a magenta rectangle is shown on the left side of the panel. Cell shapes are represented with white lines in the images corresponding to DAPI. **B** Quantification of average cell compactness in DMSO and Rockout-treated embryos. **C** Quantification of the average number of filopodia in DMSO and Rockout-treated embryos. **D** Quantification of average nuclei flatness in DMSO and Rockout-treated embryos. Detailed plot in supplementary figure 6. Boxes represent the quartiles; the whiskers indicate the maximum and minimum values. Red and black lines indicate the median and the mean, respectively. Points indicate independent embryos. $n = 7$ DMSO treated embryos examined in medial regions, $n = 7$ DMSO treated embryos examined in lateral regions, $n = 8$ Rockout-treated embryos. *P* values are indicated in the figure. Two-sided Student's *t* tests were performed to evaluate statistical significance. Scale bars = 5 μm. Source data are provided as a Source Data file.

mechanically activates Yap/Taz[16,39,40]. When this pathway is inhibited, there is a reduction of stress fiber formation and FA maturation, which translates into reduced intracellular tension[22,41]. To test whether a mechanical feedback loop is operating in migrating cells, we applied the pharmacological inhibitor Rockout to interfere with the Rho/Rock pathway. Gastrulating embryos were treated with Rockout for a short developmental window (2 h), after which cellular and nuclear morphologies, as well as Yap activation, were examined (Figs. 7, 8, S6, S8B). Rockout-treated embryos exhibited a similar phenotype to *yap* mutants, with lack of axis condensation and delayed epiboly when compared to a wild-type embryo at an equivalent 17 stage (Fig S8A). To investigate the morphology of the cells, we injected *Utrophin:GFP* together with *Lyn-tdTomato (LynTm)* mRNA at the one-cell stage in

control (DMSO) or Rockout-treated embryos. In control embryos, we confirmed that, whereas compact cells at the midline display round nuclei, lateral converging cells have a spread morphology with more filopodial protrusions and flattened nuclei (Fig. 7A–D; S6B, F, G). In contrast, cell and nuclei morphologies were significantly rounder for lateral cells in Rockout-treated embryos, as determined by their compactness and nuclei flatness and sphericity indexes, as well as a reduced number of cell filopodia; all suggesting a reduced intracellular tension (Fig. 7A–D; S6B, F, G; Supplementary Movie 5). To determine if Rock inhibition impinges on Yap transcriptional activity, we followed *marcksl1b* expression, a potential direct target of Yap according to our data (Figs. 3, 4), as well as the activation of the Tead/Yap sensor.

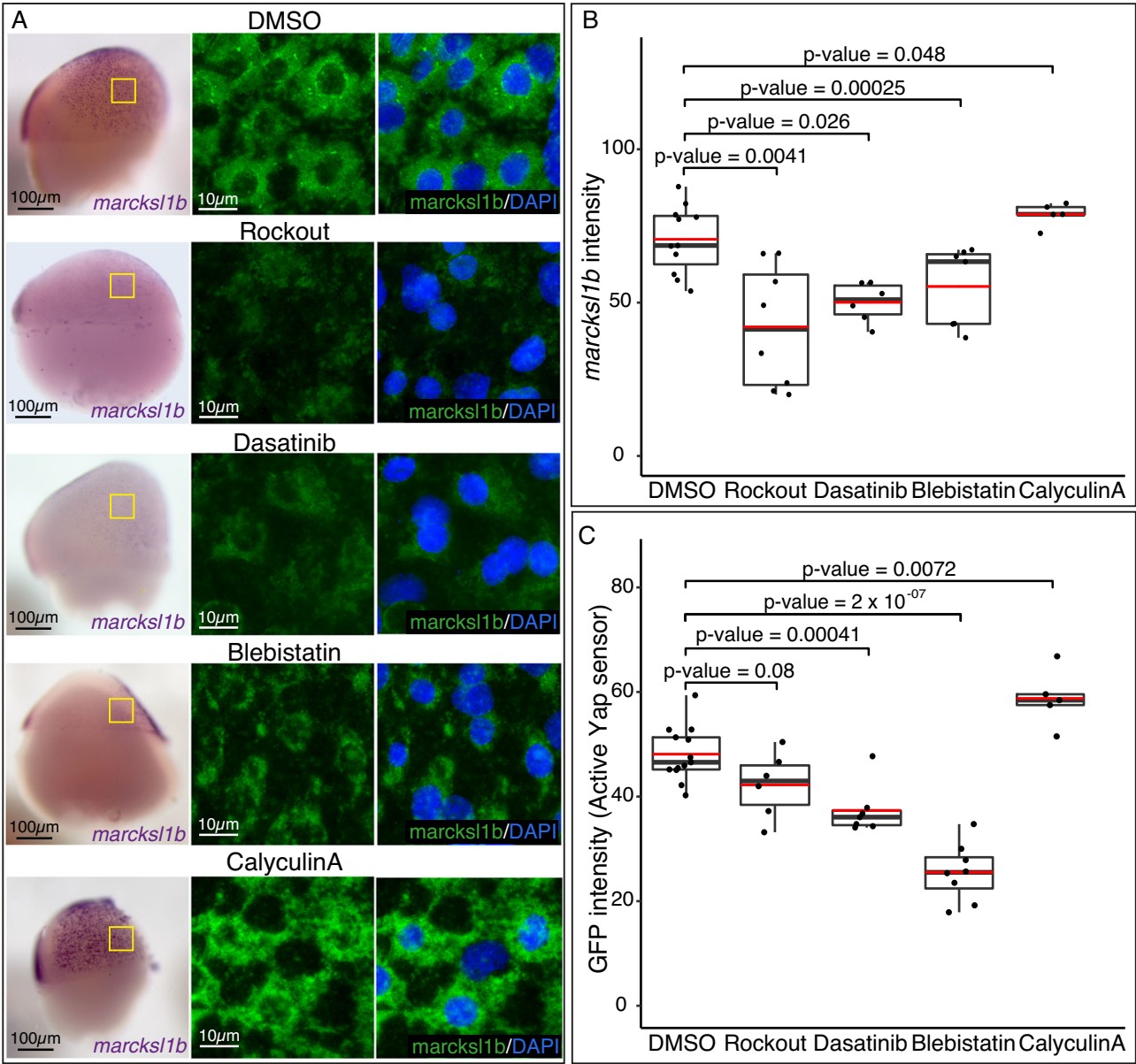

**Fig. 8 | Yap activity response to intracellular tension alterations. A** ISH analysis distribution of *marcksl1b* in WT embryos treated with DMSO, Rockout, Dasatinib, Blebistatin, or CalyculinA for 2 h. Lateral stereo microscope images of stage 16 embryos are shown (first row). Confocal microscopy images of *marcksl1b* fluorescent ISH stained with DAPI (second and third rows) from the areas indicated with yellow rectangles are shown. **B** Quantification of *marcksl1b* fluorescent ISH signal in DMSO, Rockout, Dasatinib, Blebistatin, or CalyculinA treated embryos. *P* values are indicated in the figure. *n* = 11 DMSO treated embryos, *n* = 8 Rockout-treated embryos, *n* = 6 Dasatinib-treated embryos, *n* = 7 Blebistatin treated embryos, *n* = 5 CalyculinA treated embryos. **C** Quantification of GFP signal intensity in DMSO,

Rockout, Dasatinib, Blebistatin, or CalyculinA treated transgenic embryos for the Tead/Yap sensor *GTIIC::GFP*. *P* values are indicated in the figure. *n* = 13 DMSO treated embryos, *n* = 6 Rockout-treated embryos, *n* = 7 Dasatinib-treated embryos, *n* = 8 Blebistatin treted embryos, *n* = 5 CalyculinA treated embryos. Boxes represent the quartiles; the whiskers indicate the maximum and minimum values. Red and black lines indicate the median and the mean, respectively. Points indicate independent embryos. Two-sided Student's *t* tests were performed to evaluate statistical significance. Scale bars = 100 μm and 10 μm (**A**). Source data are provided as a Source Data file.

Quantitative imaging analysis revealed a significant reduction in *marcksl1b* expression and Tead/Yap activity in Rockout-treated embryos when compared to WT, thus indicating a diminished transcriptional activation by Yap (Fig. 8A–C, S8B). Next, to further confirm the connection between tension and Yap activity in gastrulating cells, we applied alternative pharmacological treatments such as Blebblistatin, a selective myosin II inhibitor[42], and Dasatinib, a kinase inhibitor active against Src family kinases that interferes with both FAs dynamics and the Src/Yap signaling axis[43,44]. Similar to Rockout-, Blebblistatin- and Dasatinib-treated embryos, mimicked *yap* mutants' phenotype and displayed a marked reduction in *marcksl1b* expression

and Tead/Yap activity (Fig. 8A–C, S8B). Finally, to close the loop, we investigated if an increase in the intracellular tension will as well translate in an increment of Yap nuclearization and activation of its transcriptional program. We followed two different approaches; first, a pharmacological activation of Myosin II using CalyculinA[45], and second, a direct mechanical stimulation of the embryos by using a customized mechanical tester (Univert; CellScale) for simultaneous uniaxial compression of gastrulating embryos (Fig S9A). CalyculinA treatment resulted indeed in a Yap over-activation in gastrulating embryos, as indicated by an enhanced *marcksl1b* expression and Tead reporter activity (Fig. 8A–C, S8B). Interestingly, the mechanical compression of

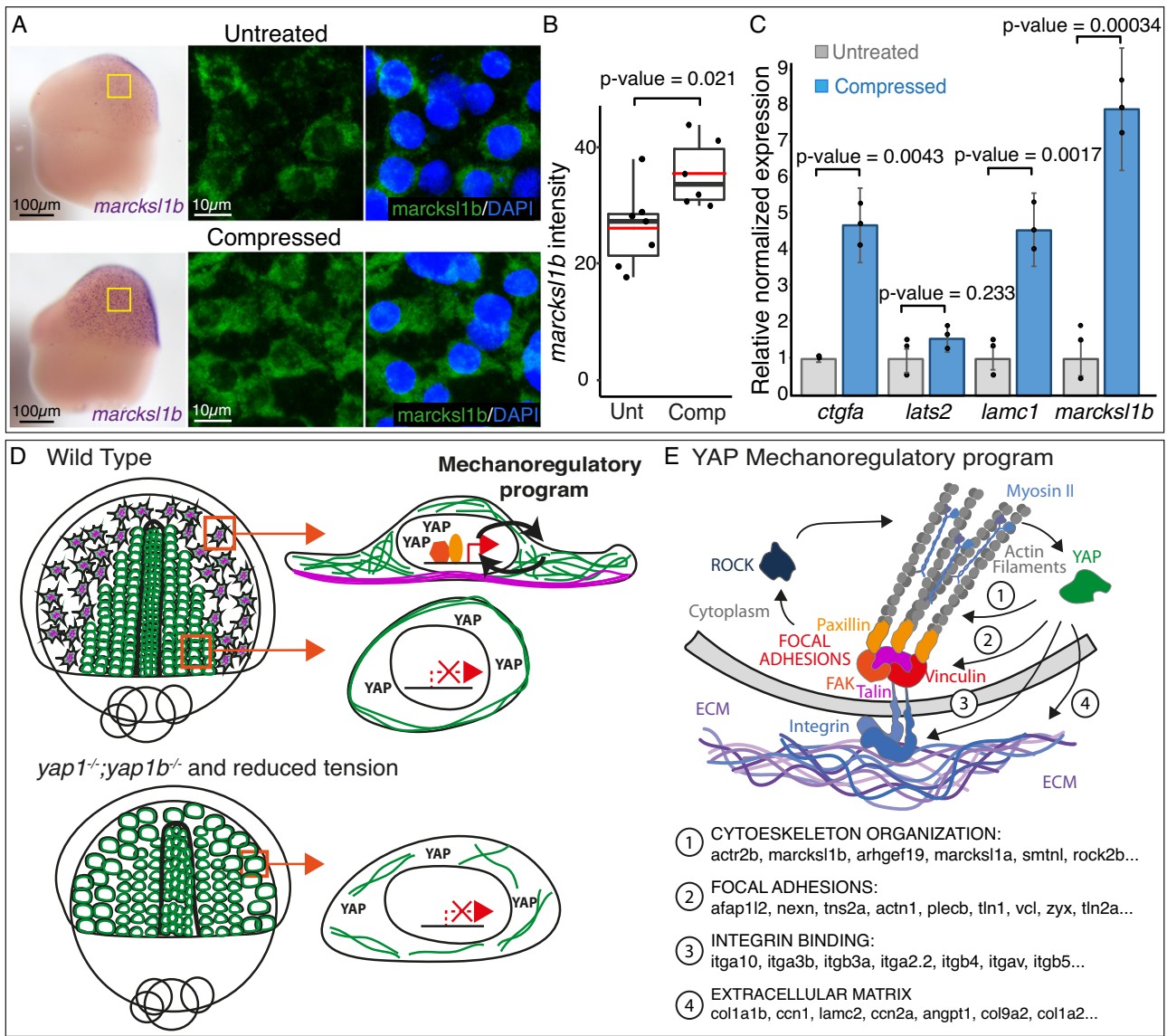

**Fig. 9 | Yap senses intracellular tension within a mechanoregulatory feedback loop. A** ISH analysis distribution of *marcksl1b* in control (untreated) and compressed WT embryos during 20 min. Lateral binocular images are shown. Confocal microscopy images of *marcksl1b* fluorescent ISH stained with DAPI from the sections indicated with yellow rectangles are shown. **B** Quantification of *marcksl1b* fluorescent ISH signal intensity in control and compressed embryos. *P* value = 0.021. Boxes represent the quartiles; the whiskers indicate the maximum and minimum values. Red and black lines indicate the median and the mean, respectively. Points indicate independent embryos. *n* = 7 control embryos, *n* = 6 compressed embryos. **C** mRNA levels of *ctgfa*, *lats2*, *lamc1*, and *marcksl1b* in control and compressed embryos as quantified by RT-qPCR. *P* values are indicated in the figure. Data are represented as mean ± SD; points indicate technical replicates. *n* = 20 embryos. **D** Summarizing scheme representing the differences between medial and lateral migrating cells converging to the midline in WT, *yap1−/−;yap1b−/−* and reduced tension embryos. **E** Main components of the Yap-dependent transcriptional program encode for proteins that provide a link between the ECM and the actin cytoskeleton. Two-sided Student's *t* tests were performed to evaluate statistical significance. Scale bars are 100 μm and 10 μm (**A**). Source data are provided as a Source Data file.

the embryos for a short time window (20% axial deformation for 20 minutes) also produced a similar increase in *marcksl1b* expression (Fig. 9A, B) following direct image quantification. In addition to *marcksl1b*, we also assessed the expression of other Yap bona-fide targets, such as *lamc1*, *ctgfa*, and *lats2* by quantitative PCR (qPCR). This analysis showed a significant increment of the expression of the Yap targets *marcksl1b*, *lamc1*, and *ctgfa* after embryo compression (Fig. 9C). Finally, we checked the effect of mechanical compression in embryos after pharmacological inhibition by Rockout treatment. We saw that in Rock inhibited embryos, the impact of the mechanical perturbation on the Yap-dependent transcriptional activation is largely reduced (Fig S9). This observation is in agreement with our previous findings (Fig. 8), and most probably indicates that Yap ability to

mechanotransduce relies on the generation of intracellular tension. Taken together, our data point to the existence of a mechanoregulatory loop between intracellular tension and Yap activation, which is essential to maintain directed cell migration during gastrulation (Fig. 9D).

## Discussion

### Yap is specifically active in dorsally migrating cells towards the midline

Identifying the underlying cues of cell migration is key, both to understand self-organization principles behind tissue assembly and homeostasis, as well as to identify molecular targets to fight malignant metastasis. Here, we uncover the role of Yap as a transcriptional hub to

coordinate the genetic program required for directed cell migration during gastrulation. The formation of the embryo axis in teleosts entails cells gathering at the midline through different morphogenetic processes that vary depending on their position. While ventral and anterior cells, closer to the embryo midline, converge following mainly radial and mediolateral intercalation movements, lateral cells away from the midline converge via directed cell migration[1,46]. We showed that Yap is active just in these lateral cells that are far from the midline and need to undergo long displacement. In contrast, cells closer to the midline would converge in a Yap-independent manner, being Wnt/planar cell polarity (PCP)-dependent mediolateral intercalation the most likely responsible mechanism[47–49]. We observed a significant correlation between Yap activity silencing and increased cell density closer to the embryo midline and upon yolk/embryo size reduction. Therefore, we propose that the shape changes imposed by the spatial restrictions in more crowded areas (e.g., cells become rounded, reducing their ECM contact surface and protrusions) might facilitate Yap inhibition, as previously demonstrated[14,36]. Aragona et al. reported that the main determinant for Yap/Taz inhibition is actually to accommodate to a smaller cell size. Small cells attach to a smaller ECM substrate area, displaying "decreased integrin-mediated focal adhesions, reduced actin stress fibers, and blunted cell contractility"[14]. Thus, culturing cells in soft substrates, placing them in suspension, or disrupting the F-actin cytoskeleton are all scenarios that result in Yap/Taz signaling inhibition[14,50–52]. In the opposite cell configuration (i.e., spreading cells), mechanical cues have been shown to be essential for Yap/Taz nuclear localization and activity[13,52]. The work of Elosegui-Artola et al. illustrates how cytoskeletal and cell shape changes affect Yap activation. They describe how cell flattening triggers nuclear pores relaxation, allowing transcription factors, such as Yap, to enter the nucleus upon cell deformation[16]. Our results show that Yap-activated cells show the typical spreading shape, with more flattened nuclei, and consistently *yap* mutant cells display a more spherical morphology. As these Yap-activated cells approach the midline region, the spatial restrictions imposed by a higher cell density would lead to a rounder nuclei conformation, thus decreasing Yap nuclear activation.

## Yap coordinates a morphogenetic program by activating actin cytoskeleton regulators, ECM, and focal adhesion components

Yap proteins function in a broad range of processes (i.e., the control of cell fate, proliferation, apoptosis, and movement), and thus, have a key role in development, tissue remodeling, and tumor progression[17,18]. However, in the context of gastrulation, we showed that Yap paralogs play almost exclusively a morphogenetic role. In contrast to previous findings at later stages of development[19,23], we did not observe a Yap involvement in cell proliferation and cell death at gastrulation stages. Moreover, our RNA-seq and ISH data also indicate that, in general terms, germ layers specification (i.e., endoderm, mesoderm, and neuroectoderm) is not affected in *yap* double mutants. The exception to this is the specification of the non-neural/epidermal lineage. We showed that key non-neural specifiers, such as *dlx3b*, *gata2a* and *tp63* appear downregulated in double mutant embryos. This finding is in agreement with recent reports indicating a key role for Yap/Tead in the determination of the non-neural/epidermal lineage[53,54]. However, it is very unlikely that epidermal lineage misspecification is behind the strong axial defects observed in *yap* double mutants, as the mutation of key non-neural/epidermal specifiers, such as *tp63* do not result in gastrulation defects in teleosts[54]. The requirement of Yap in fate specification remains controversial in hESCs cultures, as Yap-dependent ectodermal differentiation has been reported only when the ectoderm lineage is induced in hESCs-derived 2D gastruloids[55,56].

Our bulk RNA-seq results suggest a tight relationship between Yap activation and an increase in the expression of genes mostly involved in cell-ECM adhesion, cytoskeleton organization, and cell migration. Furthermore, a significant proportion of these identified genes are putative direct targets of Yap1/Yap1b, according to our previous DamID-seq datasets[23]. This suggests a straightforward regulatory role in cell adhesion and cell migration during gastrulation. A similar transcriptional program, coupling intrinsic cell tension with adhesion to the ECM, has also been reported in migratory endothelial and breast cancer cells[22,57]. Among genes activated by Yap in our study, we can identify a number of them with a previously identified role in gastrulation, particularly in controlling convergent-extension movements in zebrafish. This is the case of genes encoding for integrins[58] and ECM molecules[59], as well as cytoskeletal regulators such as members of the *marcks* family[60], *akap12b*[61], *rock2b*[62] or *vangl2*[63]. Besides these gastrulation-related genes, the transcriptional program controlled by Yap comprises a list of common 'beacon' genes, which have been proved to be direct transcriptional targets of Yap/Tead complexes in previous studies, regardless of the cell type and developmental stage considered. This list includes genes such as *ctgfa*, *cyr61*, *amotl2b*, *lats2*, or *col1a1b*[23,28,64]. In agreement with this, despite the different cellular context and associated phenotypes, this gene battery, identified as downregulated in *yap1/yap1b* medaka mutants, was also found as differentially expressed in *yap1/wwtr1* zebrafish mutants[65] (e.g., *amotl2b*, *cyr61*, *cdc42ep*, *sorbs3*, *ctgfa*, *col1a1b*, and *pcdh7*). In this study, the authors analyzed the role of *yap1* and its paralog *wwtr1* (a.k.a. *taz*) in zebrafish embryo development. They reported that double mutant embryos show a defect in the elongation of the posterior part of the embryo, by regulating the deposition of Fibronectin (a main ECM component) in the presumptive epidermis[65]. The main difference with our results is that the phenotype observed in zebrafish was first evident at 15, 16 somite stage, much after gastrulation was completed, and did not interfere with the assembly of the primary embryo axis. Similarly, in mice, axial elongation defects are observed after gastrulation in *Yap1* mutant embryos[66]. Given the mild axial defects reported in mice and zebrafish mutants, it would be premature to rule out an early role for Yap proteins in controlling gastrulation rearrangements in vertebrates. A logic assumption is that Yap signaling cooperates with other mechanisms to direct cell migration, and thus it is possible that its early role has remained elusive due to compensatory mechanisms. Additionally, it is important to take into account that the spatial configuration of the gastrulating cells varies among species. Due to the much larger size of the yolk, dorsally converging cells in medaka embryos are more flattened and have to travel longer distances to arrive at the midline compared to their equivalent in zebrafish. Therefore, it is likely that the particular geometry of the medaka gastrula has facilitated uncovering the role of Yap proteins in directed cell migration.

## Yap senses and maintains tension within a mechanoregulatory loop

The way that, within tissues, forces are generated, sensed, and transmitted is increasingly being understood as a continuous interplay between the cells and their environment. This results in regulatory feedback loops in which cells perceive mechanical cues and respond in turn modifying its own mechanical properties[7,67]. In the case of Yap, the general agreement is that the cytoskeletal organization reflects the mechanical state of the tissue and serves as a universal Yap activator; while Yap will transform these inputs into transcriptional changes inducing more cytoskeletal rearrangements[40]. A general role of Yap in maintaining tissue integrity and shape during organogenesis has been postulated[19]. Here, our results suggest that, in the context of gastrulation, Yap plays a pivotal role in the establishment of a mechanoregulatory program (Fig. 9D). We show that intracellular cell tension activates Yap, which in turn triggers a genetic program that will contribute to sustain tension levels by maintaining the spreading shape of gastrulating cells. This feedback points to the existence of a mechanoregulatory loop between intracellular tension and YAP transcriptional program activation in migratory cells. Yap-dependent feedback mechanisms have been described in diverse cellular contexts, ranging

from cardiomyocyte regeneration[68], breast cancer cells[69], mesenchymal stem cell cultures[22], and endothelial cells migration[57]. What our results and all these examples have in common is that mechanically activated Yap/Taz promotes F-actin remodeling, FA assembly, and integrin and ECM components expression; all essential elements mediating ECM-cell communication[40]. These feedback loops confer robustness to the morphogenetic processes and integrate them into their organismal context. An alternative to our proposed YAP-dependent mechanoregulatory loop in migratory cells is that Yap has a generic role in maintaining tissue integrity in the entire embryo[19], and in consequence just a permissive function in cell migration. However, we observed that YAP is specifically active in migratory cells at gastrulation stages and shuts down at the embryo midline, where cells are densely packed and do not longer move dorsally. Therefore, our results favor the scenario where Yap plays a specific function in migratory precursors to maintain their distinct geometrical and mechanical features.

Here we have identified Yap as a transcriptional hub that orchestrates the cytoskeletal changes required for a cell to move long distances and arrive on time to their final destination during gastrulation.

## Methods

### Strains and fish maintenance

The medaka (*Oryzias latipes*) iCab wild-type strains, the transgenic lines *tg(4xGTIIc:eGFP)*, and the mutant strains *yap1Δ7pb* and *yap1bΔ136pb* were maintained under previously described experimental conditions[23]. To generate the medaka line *tg(4xGTIIc:eGFP)*, the plasmid *4xGTIIc:eGFP* with flanking Tol2 sites[70] was injected at 10 ng/μl together with in vitro transcribed Tol2 RNA (50 ng/μl) into one-cell-stage iCab embryos. GFP-positive embryos were raised to adulthood and outcrossed to iCab WT fish to establish the transgenic line *tg(4xGTIIc:eGFP)*.

Animal experiments were carried out according to ethical regulations. Experimental protocols have been approved by the Animal Experimentation Ethics Committees at the Pablo de Olavide University and CSIC (license number 02/04/2018/041).

### 3D reconstruction of *yap1/yap1b* mutant embryos

Wild-type, *yap1* single mutant, and *yap1/yap1b* double mutant siblings at stage 17 were fixed with PFA 4% at 4 °C for 2–3 days. Samples were washed extensively in PBS-0.2% Tween and stained with phalloidin Alexa-488 (Invitrogen) in PBS-0.2% Tween solution supplemented with 5% DMSO (1:50) overnight (o/n) at 4 °C. After extensive washing steps with PBS-0.1% Tween, samples were stained with DAPI (1:1000), mounted in FluoroDish 35 mm plates (WPI) and imaged in a Leica SP5 microscope using a ×20 multi-immersion objective. Embryos were imaged dorsoventrally taking images of 50 stacks of 3 μm-length each with a pixel size of $0.379 \times 0.379$ μm$^2$. Tridimensional models were acquired using Imaris v8.02. After imaging the embryos, each one of them was individually genotyped.

### Whole-mount embryo immunostaining

Embryos collected from *yap1$^{+/-}$;yap1b$^{+/-}$* adult fishes were fixed at stage 16 using 4% PFA. Fixed embryos were dechorionated with forceps. Embryos were washed with PBS-0.2% Tween, treated with cold acetone at −20 °C for 20 min, then incubated with freshly prepared blocking solution (10% fetal bovine serum in PBS-0.2% Tween) at room temperature (RT) for 2 h. The primary antibodies anti-active caspase-3 antibody (BD Biosciences, 559565) and anti-phospho-Histone H3 (Ser10) antibody (Millipore 06-570) were diluted 1:500 in blocking solution and embryos were incubated o/n at 4 °C. Embryos were then subsequently washed with PBS-0.2% Tween and incubated o/n at 4 °C in the dark with the Alexa Fluor TM 555 Goat anti-rabbit antibody (Invitrogen #A32727), diluted as well 1:500 in blocking solution.

Finally, embryos were washed with PBS-0.2% Tween and incubated o/n at 4 °C with DAPI (Sigma) diluted 1:1000 in PBS-0.2% Tween. For imaging, embryos were embedded in 1% low-melting-point agarose and mounted in FluoroDish 35 mm plates. Confocal laser scanning microscopy was performed using a Zeiss LSM 880 microscope. Images were processed using ImageJ v3.96.3/v65[71]. For quantification of apoptotic and proliferative cells, masks were applied for both channels. The mask generated for the red channel was segmented using the Watershed algorithm. Only the regions marked with the primary antibody that also corresponded to nuclei with a 6–200 μm$^2$ area were considered to avoid debris. Apoptotic and proliferative cells were counted on the embryo surface and extrapolated to the total number of nuclei (quantified as described above using the corresponding DAPI images). After imaging, embryos were genotyped by PCR to identify *yap1/yap1b*-related genotypes. To analyze whether experimental groups were significantly different, two-sided Student's *t* tests were performed.

### Analysis of cell movements during gastrulation

WT, *yap1* single mutant, and *yap1/yap1b* double mutant siblings were injected at one-cell stage with H2B-GFP mRNA (Addgene, #53744) at a final concentration of 25 ng/μL. The embryos were incubated for 3 h at 28 °C and o/n at 25 °C. The most promising candidates were then selected the day after in a fluorescent binocular and dechorionated following a three-step protocol with minor modifications[72]. First, embryos were rolled in sandpaper (2000 grit size, waterproof) to weaken their outer structure. Then, the embryos were incubated for 30 min at 28 °C in pronase at 20 mg/ml. Finally, after several washing steps, embryos were incubated for 60 min in hatching enzyme at 28 °C and transferred into a Petri dish with BSS 1× medium supplemented with penicillin-streptomycin and 1-heptanol 3.5 mM (Sigma) to block contractile rhythmical movements. Overnight movies (8–9 h) were acquired using a Leica SP5 microscope with a ×20 objective. Frames with a pixel size of $0.189 \times 0.189$ μm$^2$ were taken every four minutes. Each frame, 2 confocal section separated 10 μm were acquired, maximum projected, and processed with an unsharp mask (radius sigma = 15 and mask weight = 0.60) and median filter (radius = 2.0). Manual cell-tracking analysis was carried out using ImageJ v3.96.3[71]. Alternatively, a more precise, semi-automatic cell-tracking was also performed using TrackMate v6.0.1/v65[73] (blob diameter = 7.9 and threshold = 0.23). The resulting data were analyzed using R. Cells tracked in less than 15 frames and/or localized initially near the midline were excluded. Lateral or midline cell trajectories were determined according to their central position with respect to the midline. Variance test and two-sided Student's *t* tests were performed to estimate the statistical significance among the different experimental conditions.

### RNA-seq

**Library preparation.** Individual wild-type, *yap1* single mutant, and *yap1/yap1b* double mutant embryos at stage 16 were homogenized in TRIzol (Ambion). Samples were centrifuged at full speed and the supernatant was transferred to a fresh tube. Chloroform was then added to split the RNA (upper aqueous phase) from the DNA fraction (lower phase). The DNA fraction was precipitated by adding glycogen and 100% ethanol and incubating it at RT for 20 min. This fraction was then centrifuged at maximum speed for 30 min at 4 °C. After three washing steps using 75% ethanol, the DNA pellet was then resuspended in 30 μL of TE buffer. The RNA fraction was precipitated by adding RNA-grade glycogen (ThermoFisher Scientific) and isopropanol and following the same steps applied to the DNA fraction. The RNA pellet was resuspended in 12 μL of nuclease-free water. Each embryo was genotyped using its corresponding purified DNA fraction. The RNA samples were merged according to their genotype to generate from three to four biological replicates. Prior to library preparation,

contaminating DNA remnants were degraded using the *TURBO DNA-free kit* (Ambion). Each RNA library was finally sequenced using an Illumina HiSeq 2500 system.

**Downstream bioinformatic analysis.** Reads were pre-processed trimming the Illumina universal adapters and the first 12 bases of each read to avoid k-mers using *Trimmomatic* v0.39[74]. Reads shorter than 50 bp and those with an average quality lower than 20 were filtered out (HEADCROP:12 MINLEN:50 AVGQUAL:20). Potential rRNA sequences were removed using *sortmerna* v2.1[75]. Processed reads were then mapped against the last version of the medaka genome (ASM223467) using Hisat2 v2.1[76]. Only reads with a high mapping quality (*samtools* v0.1.19-96b5f2294a *view -q 60*) were considered for further steps of the bioinformatics analysis. The software *htseq-count* was used to count the number of reads per gene (GTF from Ensembl version 99 was used as a reference). The subsequent analysis was performed using DEBrowser v1.14.2[77]. Genes with less than 10 reads on average were discarded (RowMeans <10) and data were normalized following the relative log-expression (RLE) method. Potential batch effects were removed using ComBat (included in DEBrowser v1.14.2). Differential gene expression analysis was carried out using the R package DESeq2 (included in DEBrowser v1.14.2) (*p*adj <0.05; −log FC = 1; https://www.r-project.org/).

GO Terms were analyzed using GProfiler ve108_eg55_p17_39cdea3[78]. The lists of *yap1* and *yap1b* DamID peaks obtained previously[23] were concatenated and, after assigning the closest gene using *Bedtools* v.2.21.0, they were compared to the list of downregulated genes in our bulk RNA-seq to identify which genes are potential direct binding targets for Yap1 and Yap1b. As the Ensembl version used for the RNA-seq analysis is more recent than the one used for the DamID-seq analysis (Ensembl version 89), those genes which may have changed their identifier were not considered for this analysis. The statistical significance for the comparison between control and identified overlapping DamID peaks was calculated applying a two-proportion Z-test.

**Whole-mount in situ hybridization**
cDNA from medaka embryos at stage 24 was used to amplify part of the coding sequence of medaka *no-tail* (ntl), *goosecoid* (gsc), *sox3*, *yap1*, and *marcksl1b* genes. PCR products were cloned into pSC-A-amp/kan Stratagene plasmids (Agilent) to generate probes for whole-mount in situ hybridization (ISH) experiments (Supplementary Table 2). Probes were synthesized using digoxigenin-11-UTP nucleotides (Roche) and the T3 or the T7 polymerase (Roche) depending on the insert orientation. Probes generated were used in a final concentration of a 3 ng/µl. ISH was performed following a previous protocol[79]. Medaka embryos at stage 15, 16, 17, and 18 were fixed in 4% PFA for two days, dehydrated in methanol, and stored at −20 °C.

Fluorescent in situ hybridization (FISH) was performed on medaka embryos at stage 16 using specific probes for *marcksl1b*. We followed the same protocol as for ISH, with the following modifications from incubation with the anti-DIG antibody on: samples were first Incubated with blocking Buffer (2% Blocking Reagent from ROCHE in MABTween 1x) for 1 h, and then with anti-digoxigenin-POD antibody (11207733910 Roche, 1:150 in Blocking Buffer) for at least 2 h at RT. The embryos were then washed six times with PBS 0.1% Tween at RT and then o/n at 4 °C. Later, the embryos were washed again with PBS-0.1% Tween and three times with Borate buffer (100 mM Borate Buffer, 0.1%Tween), and stained with TSA amplification solution (50 µg/ml TSA Fluorescein 5 mg/ml in 100 mM Borate Buffer, 0.1%Tw, 2% DS, 0.003% H2O2) for 1 h at RT in the dark. Finally, embryos were incubated overnight at 4 °C with DAPI (Sigma) diluted 1:1000 in PBS-0.2% Tween. Stained embryos were mounted in FluoroDish plates as described previously. Confocal laser scanning microscopy was performed using a Zeiss LSM 880 microscope with a 40x objective. Only embryos that were mounted

with their dorsal-anterior axis oriented in parallel to the cover glass bottom were used for the analysis. Maximum projection images were processed using ImageJ v3.96.3/v65[71]. A list of primers used to generate the RNA probes is provided in Supplementary Table 2.

To quantify marcksl1b expression, mean gray values of the GFP channel were obtained from 56 µm² regions of maximum projection images using ImageJ. To establish a correlation between m*arcksl1b* and cell density, we determined the centroid position of each nucleus using TrackMate v6.0.1, excluding the centroids that were closer than eight pixels and those nuclei at the border of the image. Then, we calculated the mean of the distance between the five closest neighbors for each nucleus. Based on the *marcksl1b* expression pattern, we distinguished two different areas; an active area which shows a specific expression pattern for *marcksl1b* and an inactive area in which cells are not expressing *marcksl1b*. We represented the XY position of the nuclei's centroids, with a circle or a triangle if they were localized in the active or the inactive area, respectively. The color gradient of each nucleus was dependent on the mean distance to its five closest neighbors. To analyze whether experimental groups were significantly different, two-sided Student's *t* tests were performed.

**mRNA Generation and injection**
DNA plasmids containing *yap1::mcherry, utrophin::GFP*[80], *paxillin::mKate*[81] (Addgene 105974), *and lynTdTomato* were linearized with NotI and then transcribed using the mMESSAGE mMACHINE SP6 Kit (Ambion) to synthesize capped mRNA. RNA was injected into one-cell-stage embryos; *yap1::mcherry* (80 pg/embryo), *utrophin::GFP* (150 pg/embryo), *paxillin::mKate* (125 pg/embryo), and l*ynTdTomato* (100 pg/embryo). For the phenotypic rescue experiments of *yap1−/−;yap1b−/−* embryos with Paxillin, we injected 300 pg per embryo of *paxillin::mKate* mRNA.

Embryos injected with *utrophin*::GFP and paxillin::*mKate* or *lynTdTomato* mRNA were fixed in PFA 4% at stage 16 for two days. Fixed embryos were washed with PBS-0.1% Tween, dechorionated with forceps, and incubated overnight at 4 °C with DAPI (Sigma) diluted 1:1000 in PBS-0.1% Tween. Embryos were imaged with a Zeiss LSM 880 microscope (63x objective), taking images of a pixel size of 0.132 × 0.132 µm² and a voxel depth of 0.24 µm. Only embryos that were mounted with the dorsal-anterior axis oriented in parallel to the cover glass bottom were used for analysis. Imaged embryos were genotyped later to identify *yap1/yap1b*-related genotypes. Images were processed using ImageJ v3.96.3/v65[71]. For each image and channel, a maximum projection was generated. For XZ projections we used the Volume viewer v2.01.2 plugin from ImageJ. To analyze the 3D morphology of the nuclei we applied first a Gaussian Blur 3D filter (X, Y, and Z sigma value was set to 2.0) to the blue channel. Then, we segmented and created a 3D mask of these nuclei using plugin interactive watershed segmentation (SFC FIJI plugin 1.2.0). 3D nuclei reconstructions were performed by applying Reslice, without avoiding interpolation, and 3D Viewer v4.0.3 to DAPI signal channel. To obtain the geometrical and morphological parameters we used the plugins 3D Geometrical measure and 3D shape measure from ImageJ (included in 3D FIJI plugin mcib3d_plugins-3.96.3). Downstream analyses were carried out using R v11.453. Nuclei smaller than 100 and bigger than 350 (volume unit) were excluded.

Focal adhesions and cell morphology quantifications were performed using ImageJ v3.96.3/v65. To analyze focal adhesions, we first carried out a manual segmentation and then created a mask of paxillin::mKate signal. Then, we applied Skeletonize and measured the length of the segmented signal. To perform cell morphology analysis, we manually segmented utrophin::GFP signal, or LynTdTomato and utrophin::GFP merged signals, and measured cell area, compactness (ratio of cell area to the area of the circle having the same perimeter), and number of filopodia. Number of filopodia was defined as the number of elements whose area is larger than 0.077 µm², as obtained

from the subtraction between the total cell area and the area obtained when applying an opening morphological filter (element: disk, radius: 1.1 µm) (plugin MorphoLibJ 1.4.2.1). To analyze whether experimental groups were significantly different, two-sided Student's $t$ tests were performed.

## Yolk extraction

Removal of yolk material was performed on medaka embryos at stage 15, using microcapillaries, to obtain a 30-40% reduction of embryo perimeter. Following a 2 h incubation, embryos were fixed with PFA 4% for two days to perform ISH of marcksl1b, as described in the previous section. Yap-active area was quantified using ImageJ v3.96.3/v65[71]. To analyze cell morphology and quantify cell density, yolk removal, and fixation were also performed on embryos injected with *utrophin::GFP* mRNA. Fixed embryos were stained with DAPI (1:1000) and then imaged with a Zeiss LSM 880 microscope (objective ×40). Only embryos mounted in such a way that their dorsal-anterior axis was oriented in parallel to the cover glass bottom were used for the analysis. Maximum projection images were processed using ImageJ v3.96.3/v65. For cell quantification, mask and Analyze Particle were applied for the DAPI channel. Only nuclei displaying an area bigger than 7.1 µm2 were considered, to avoid including cellular debris in the measurements. To follow Yap/Tead activation, yolk extraction and fixation were performed on embryos of the transgenic line tg(4xGTIIc:eGFP). GFP signal was quantified as described in the previous section for marcksl1b expression. To analyze whether experimental groups were significantly different, two-sided Student's $t$ tests were performed.

## Drug treatments

Medaka embryos were dechorionated in vivo following a previous protocol[72] with minor modifications. First, embryos were rolled in sandpaper (2000 grit size, waterproof) to weaken their outer structure. Then, the embryos were incubated for 30 min at 28 °C in pronase at 20 mg/ml. Finally, after several washing steps, embryos were incubated for 60 min in hatching enzyme at 28 °C and transferred into a Petri dish with BSS 1× medium. Embryos at stage 15 were incubated with DMSO, Rho Kinase Inhibitor III (555553, Merck) at 250 µM, Myosin II ATPase inhibitor (Blebbistatin, 203391, Merck) at 300 µM, tyrosin kinase inhibitor (Dasatinib, SML2589, Merck) at 150 µM, or Phosphatases types 1 and 2 A inhibitor (Calyculin, C5552, Merck) at 0.7 µM dissolved in water for 2 h. Then embryos were fixed with PFA 4% for two days to perform ISH and FISH as described in the previous section. Embryos of the transgenic line tg(4xGTIIc:eGFP) were also treated with the above-mentioned drugs, and their GFP signal quantified as previously described. Embryos injected with utrophin::GFP and lynTdTomato mRNA were also treated with DMSO or Rho Kinase Inhibitor and fixed. Fixed embryos were then imaged with a Zeiss LSM 880 microscope and their nuclear morphology analyzed as indicated in the previous section. To analyze whether experimental groups were significantly different, two-sided Student's $t$ tests were performed.

## Embryos compression

For each condition, batches of 25 medaka dechorionated embryos were subjected to confined compression using a Univert device, equipped with a customized compression chamber (CellScale, Waterloo, ON; Figure S9A). Embryos, placed in the compression chamber, already filled E3 medium (9 ml), were mechanically stimulated by a motorized vertical indentor up to the configured axial distance (Fig S9A). A rubber ring placed between the indentor piece and the samples' chamber controlled the compression fit and precision (±1 µm) (Figure S9A). Embryo batches were compressed to 80% of their diameter, which corresponds to a uniaxial displacement of 250 µm, for a period of 20 min. Non-compressed control embryos were confined in parallel in E3 medium.

## qPCR

To measure gene expression levels after compression, mRNA from medaka embryos was isolated using easy-BLUE Total RNA Extraction Kit (iNtRON Biotechnology, Inc. Korea). Then, cDNA retrotranscription was performed using the iScript cDNA Synthesis kit (Bio-Rad). Afterwards, the concentration was measured in a Qubit fluorometer. The expression levels of *ctgfa*, *lats2*, *lamc1*, and *marcksl1b* were quantified by RT-qPCR (CFX96 Touch Real-Time PCR Detection System), normalizing the results with the housekeeping gene *ef1a*. All qPCR reactions were performed in triplicate with SsoAdvanced Universal SYBR Green Supermix in a total volume of 10 µL (Primers sequences listed in supplementary table 2). To analyze significant differences, two-sided Student's $t$ tests were performed.

## Statistics and reproducibility

The tests performed in R v11.453 to assess whether experimental groups were significantly different are indicated in the figure legends.

Most experiments were carried out at least three times, and the findings of all experiments were reliably reproduced. All replicates and precise $P$ values are documented in the figures or in Supplementary Table 3.

## Reporting summary

Further information on research design is available in the Nature Portfolio Reporting Summary linked to this article.

## Data availability

RNA-seq datasets are available in the Gene Expression Omnibus (GEO) repository (https://www.ncbi.nlm.nih.gov/geo) under the following accession number: GSE201791. As a reference medaka genome (Japanese medaka HdrR; Oryzias latipes) we are using the data deposited in ENSEMBL (https://www.ensembl.org/index.html) under the accession number: ASM223467v1. Source data are provided with this paper.

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

## Acknowledgements
We thank the CABD Proteomics, Advanced Light Microscopy and Imaging, Aquatic Vertebrates, and Functional Genomics facilities for their excellent technical assistance. We thank Elena Quesada-Hernández for her critical input on the manuscript. This work was supported by grants awarded to JRMM from the Spanish Ministry of Science, Innovation, and Universities (AEI): (References BFU2017-86339P, RED2018-102553-T, PID2020-112566GB-I00, and CEX2020-001088-M), and by the Marie Sklodowska-Curie H2020-MSCA-IF- 2018-ST MechaPattern 834610 and La Caixa Junior Leader Incoming from Fundación "la Caixa" awarded to M.A.C.

## Author contributions
A.S.O. and J.V.M. conducted most experiments, performed bioinformatic analyses, and had a main contribution in figures and manuscript editions. E.S.R. and R.P. contribute to the phenotypic analysis of the mutants as well as in stock maintenance and genotyping. J.C. performed and analyzed the embryo compression experiments. A.C.L. contributed to the quantitative analysis of cell displacements and nuclear morphologies. L.B. participated in the transcriptomic characterization of single and double mutants. F.L. designed and developed the 4xGTIIC::GFP line. J.R.M.M. and M.A.C. conceived the project and assisted A.S.O. and J.V.M. in data analysis. The manuscript was edited and written by A.S.O., J.V.M., M.A.C., and J.R.M.M. M.A.C. and J.R.M.M. co-supervised all the work.

## Competing interests
The authors declare no competing interests.
