## [Peer Review File · Nature Communications]

A Yap-dependent mechanoregulatory program sustains cell migration for embryo axis assemblyREVIEWER COMMENTS

Reviewer #1 (Remarks to the Author):

Manuscript « A Yap-dependent transcriptional program directs cell migration for embryo axis Assembly » by Ana Sousa-Ortega, Juan R. Martinez-Morales and collaborators deals with the role of the Yap pathway in medaka embryos axis assembly at gastrulation.

By performing a double knock out of Yap paralogs, the authors find strong defects in the dorsal cells migration toward the embryo mid-line in the gastrulating embryo, that results in the failure of developing axis assembly, associated to defects in direct Yap-target genes expression involved in cytoskeletal organization and cell-ECM adhesion.

They additionally find defects in the expression of the direct Yap target gene expression *marks1b* in gastrulating embryos in which cell tension has been released after pharmacological Rock inhibitor treatment.

The Yap pathway being known as mechanosensitive, the authors conclude in a hub transcriptional role of Yap in orchestrating cell migrations leading to axis assembly during Medaka gastrulation, both driving and responding to cell migration forces through a robust positive mechano-biochemical feedback loop.

This work nicely demonstrates the role of Yap as necessary for dorsal cells migration toward the mid-line leading to the embryo convergent morphogenetic movement that leads to axis assembly at gastrulation. Comprehensive analysis of gene expression show Yap-target genes expression involved in cytoskeletal organization and cell-ECM adhesion consistently implicated. The work very interestingly shows that the biochemical patterning genes involved in mesoderm versus endoderm and ectoderm specification are not implicated in the Yap dependent defects in cell migration, indicating a role of Yap as purely involved in the embryo biomechanical morphogenesis. It additionally and interestingly suggests the existence of a mechanical feedback loop based on Yap mechanosensitivity, robustly orchestrating cell migration with underlying cytoskeleton changes required for cell migration.

Major comments

1- While the authors nicely show defects in cell migration in Yap double knock-outs, it is not clear whether Yap is the motor and/or the polarizer of cell migration. Indeed, Yap could be more generically required for the expression of cytoskeletal organization and cell-ECM adhesion genes that are necessary for the full mechanical integrity of the multi-cellular tissue with other signaling pathways (downstream of embryo's biochemical patterning?) involved in directed migration. This in such a way migration defects will necessarily appear in Yap knock downs in the presence of the signaling pathways driving cell migration.

Indeed, (Porazinski, S. et al. 2015) article (1) (ref 19 of the manuscript) showed that Yap is required for the maintenance of the overall stiffness of medaka embryos to maintain its overall shape. Therefore, a role of Yap in systematically re-enforcing tissue stiffness by mechanical stimulation of the expression of genes involved in cytoskeletal organization and cell-ECM adhesion in response to morphogenetic movements regulated by other signaling pathways to avoid tissue mechanical alteration in response to forces, that would compromise biomechanical morphogenesis, would be highly interesting as well, and should be tested.

To test this alternative hypothesis, the authors should find a way to apply mechanical forces on the dorsal tissue of the embryo before migration begins, and test in immunofluorescence for the presence, concentration and the location of the major cytoskeleton and cell-ECM adhesion elements downstream of Yap, and for the integrity of the tissue in the wild type and in Yap double knock-outs.

2- The proposal of a Yap mechanical activation feedback is interesting as well, but at that stage not demonstrated into the manuscript, because no control parameter purely mechanical in nature is used

in the experiments to test this hypothesis. The existence of a mechanosensitive process necessarily requires at least one experiment involving mechanical control parameters of pure physical nature, to exclude any non-mechanical biochemical side effects due to pharmacological treatments.

To definitively test the existence of such mechanical feedback loop, the authors should again find a way to apply a mechanical deformation on the embryo, this time at cell migration stage in the presence of the rock inhibitor, and test if the rescue of a mechanical deformation from rock-inhibitor treated tension defective embryos rescues Yap activation and marcks1b expression. As well as if marcks1b expression cannot be rescued within these conditions in Yap double knock outs.

3- To exclude the existence of uncontrolled off-targets of the pharmacological inhibitor of rock, the authors should duplicate their pharmacological experiments with another inhibitor of tension, like Blebbistatin or ML7.

4- In the last paragraph of the results ("Yap senses and activates intracellular tension within a positive feedback loop"), the authors say that they check Yap activation within rock inhibited pharmacological conditions. Unless its direct target marcks1b expression is tested, the activation of Yap itself is not tested with the Tead-responsive 4xGTIIC enhancer used from the beginning of the manuscript to monitor Yap activation. The authors should test the Yap activation reporter as well.

Minor comments

1- Depending on major point 1- results, the title may be changed.

2- The species here studied, medaka, should be mentioned in the abstract.

3- Because Nature Communications is a multi-disciplinary journal with a broad scientific audience, the authors should present medaka embryos morphogenetic movements at gastrulation with the help of a scheme in their introduction.

(1) Porazinski, S. et al. YAP is essential for tissue tension to ensure vertebrate 3D body shape. Nature 521, 217-221, doi:10.1038/nature14215 (2015).

Reviewer #2 (Remarks to the Author):

This manuscript from Sousa-Ortega and colleagues addresses the role of Yap proteins in embryonic axis assembly in medaka embryos. They find that loss of both medaka yap paralogs results in a failure of the posterior body axis to form, not from defects in germ layer formation but rather from reduced migration of lateral cells toward the dorsal midline. Within wild-type embryos, Yap activity is highest in these migrating lateral cells, which adopt a flattened morphology and produce multiple actin-based protrusions to facilitate their migration. The authors report that these cells are more rounded and produce fewer protrusions in Yap double mutants, indicating a possible reduction in cortical cell tension. This is consistent with the observation that inhibition of Rho kinase phenocopies cell morphologies of Yap double mutant embryos. The function of Yap during early development, particularly its role in mechano-sensation, are likely to be of interest to a broad audience. However, while the data presented here identify several associations, further experiments are required to establish causal relationships between tissue mechanics, Yap activity, and embryonic cell migration. For this reason, I recommend the following concerns be addressed prior to publication:

1. To enhance accessibility by color-blind readers, please switch away from red-green color schemes

in figures 3, 5, 6, S4, and if possible 2D-E.

2. Please report sample (n) and trial (N) numbers for all experiments shown.

3. Are trajectories and behaviors of midline cells (where Yap is less active) also affected in Yap double mutant embryos? The authors speculate that lateral cells are primarily affected because they exhibit the highest levels of Yap activity, but without examining the behavior of Yap-low midline cells, we can't know that lateral cells are specifically or particularly affected. Further analysis of the midline cell population may provide insights into the function of low-level Yap activity, secondary effects of defective lateral cell migration, and/or cell non-autonomous roles for Yap.

4. The comparison of previously reported Yap Dam-ID seq results with differential gene expression analysis in yap mutants in figure 3 is important, but it could be reported in a more informative way. Rather than comparing only the numbers of genes shared between these datasets, I want to know what types of genes are direct Yap targets? Are any GO terms/gene sets enriched among these direct targets?

5. Figure 4 shows that the transgenic TEAD reporter is active primarily in lateral cells, while Figure S4A shows GFP within the midline cells but not lateral cells. What explains this discrepancy?

6. The authors identify a correlation between increased cell density and decreased Yap activity. This finding would be more impactful if they could establish a causal relationship. Can one experimentally increase or decrease cell density within medaka embryos to test if it modulates Yap activity?

7. The authors show that cell morphology, protrusions, and cytoskeletal organization appear different between WT and Yap double mutants, but these data would be stronger if quantified. Number/length of protrusions or number/localization of focal adhesions, for example. It would be easier to quantify the features of individual cells if the membrane marker were expressed in a mosaic fashion.

8. The authors show that inhibiting Rho kinase reduces expression of Yap target gene *marks1b*. Can this decrease in Yap activity also be seen using the live TEAD reporter? Conversely, could increasing myosin contractility using calyculin A increase Yap activity in WT embryos? Could this increased contractility rescue any aspects of the Yap mutant phenotype? Finally, ROCKOUT phenocopies some aspects of Yap mutants (such as cell flatness)... Does ROCKOUT treatment also inhibit embryonic axis assembly similar to loss of Yap?

Typographical errors:

1. Page 2, near the middle: "being dysregulated mechanical feedbacks a common landmark..."
2. Page 5, top paragraph: "... further analysis on yap double mutants most severe phenotype."

Reviewer #3 (Remarks to the Author):

In this report, Sousa-Ortega et al analysed *yap1* and *yap1b* double mutant embryos that did not properly form the body axis during gastrulation. RNA sequencing, together with the DamID data they previously reported, showed that *yap1* and *yap1b* are required for the expression of genes involved in cytoskeleton reorganization and cell-cell and cell-matrix adhesion. They next showed that Yap was active in cells migrating to the midline. In those cells, the amount of actin and the number of FAs were reduced, and their nuclei were more spherical. Finally, inhibition of Rho/Rock during gastrulation lead to rounder nuclei and a reduction in Yap-mediated transcription. They concluded that Yap activity regulates FAs and actomyosin, and that this mechanism forms a positive feedback loop.

This is a follow-up study of their previous report that provides some useful information. In their previous report, the authors identified Yap and Yap1b binding sites by using the DamID approach. They demonstrated that Yap and Yap1b have transcriptional targets controlling actin and FA formation and claimed that Yap controls mechanical properties of cells during gastrulation and forms a feedback loop. However, it has previously been shown that YAP acts as a mechano-effector from the analysis of the medaka YAP1 mutant by Porazinski et al (Porazinski et al, Nature 2015) and together with YAP's role as a mechano-transducer, they proposed the concept of a mechanical feedback loop. Furthermore, the author's claims about "direct target genes" and "feedback loop" are not fully supported by their results.

Major points:

1. In their previous report, they identified Yap1 and Yap1b binding genome sites using DamID, and suggested that Yap1 and Yap1b have similar transcriptional targets. In this report, by combining DamID data with RNA-seq data, they identified direct target genes that control the formation of actin and FAs as below. However, this conclusion is not fully supported by the results.

(1) In the abstract, they claim, "Accordingly, we identified genes involved in cytoskeletal organization and cell-ECM adhesion as "direct" Yap targets

(2) In P8, L12, the authors state, "marcks1b expression, a direct target of Yap".

(3) In P9, L28, the authors claim that "a significant proportion of these identified genes are direct target of yap1/yap1b...."

To claim this, the authors must demonstrate, (a) physical binding of Yap1 and Yap1b to the promoter/enhancer of target genes, (b) a reporter assay using the promoter/enhancer-binding sequence, and (c) a reporter assay using the mutated promoter/enhancer-binding sequence.

2. In the section titled "Yap senses and activates intracellular tension...", the authors claim to show that Yap activity regulates FAs and the actomyosin cytoskeleton, which allows for the coupling of intracellular tension to the ECM and suggest the existence of mechano-regulatory feedback. Porazinski et al. previously reported the Yap's newly discovered function as a mechano-effector in which medaka YAP mutants showed reduction of tissue tension and aberrant ECM formation. They also suggested the presence of a mechano-regulatory feedback together with Yap's previously reported mechano-transducer functions. Furthermore, the author's description that "Yap senses..." is not correct. Yap is a mechano-transducer, but it is not a mechano-sensor (Dupont et al., Nature 2011).

3. The authors drew their conclusion on YAP's control of mechanical properties based on their inference from the nuclear shape and the actin reporter Utrophin::GFP (Fig. 5, 6A, B). However, nuclear shape does not always correlate with cell compression, and actin over-polymerization leads to lower tissue tension (Pinto et al., Dev Cell, 2012). To make such a statement, they must carry out direct measurements, e.g. laser cutting of actin filaments and micro-pipette analysis.

4. While in the anterior axis, GTIIC:GFP seems not to be active (Fig.4 A), marcks1b is expressed (Fig.4 B) and depicted as the YAP+ in D (Fig.4 D). These descriptions appear to be consistent.

Minor points:

1. P2 L8: ESC (Embryonic Stem Cell)

2. P2 L22: well-known transcriptional regulators "that are" able to....

3. P3 L18: role for the mechanosensory Yap, mechano-transducer

4. P3 L36: "without an apparent definition", "formation" is better.

5. P4 L11: we asked ourselves if...

6. P5 L5: the mechanism behind Yap, "underlying" is better

7. P7 L2: Pax::mKate mRNA, needs a reference.

8. P9. L21 However, it is very unlikely... This sentence is unclear.

9. P21. Ref 20 and 23 are lacking information about the number of pages.

Manuscript: NCOMMS-22-22583A: A Yap-dependent mechano-regulatory loop directs cell migration for embryo axis assembly: Ana Sousa-Ortega ^{1*}, Javier Vázquez-Marín ^{1*}, Estefanía Sanabria-Reinoso ¹, Jorge Corbacho¹, Rocío Polvillo ¹, Alejandro Campoy-López ¹, Lorena Buono ¹, Felix Loosli ², María Almuedo-Castillo ^{1#}, Juan R. Martínez-Morales ^{1#}.

REVIEWER COMMENTS (Point by Point answer)

Reviewer #1 (Remarks to the Author):

Manuscript « A Yap-dependent transcriptional program directs cell migration for embryo axis Assembly » by Ana Sousa-Ortega, Juan R. Martinez-Morales and collaborators deals with the role of the Yap pathway in medaka embryos axis assembly at gastrulation.

By performing a double knock out of Yap paralogs, the authors find strong defects in the dorsal cells migration toward the embryo mid-line in the gastrulating embryo, that results in the failure of developing axis assembly, associated to defects in direct Yap-target genes expression involved in cytoskeletal organization and cell-ECM adhesion.

They additionally find defects in the expression of the direct Yap target gene expression *marks1b* in gastrulating embryos in which cell tension has been released after pharmacological Rock inhibitor treatment. The Yap pathway being known as mechanosensitive, the authors conclude in a hub transcriptional role of Yap in orchestrating cell migrations leading to axis assembly during Medaka gastrulation, both driving and responding to cell migration forces through a robust positive mechano-biochemical feedback loop.

This work nicely demonstrates the role of Yap as necessary for dorsal cells migration toward the mid-line leading to the embryo convergent morphogenetic movement that leads to axis assembly at gastrulation. Comprehensive analysis of gene expression show Yap-target genes expression involved in cytoskeletal organization and cell-ECM adhesion consistently implicated. The work very interestingly shows that the biochemical patterning genes involved in mesoderm versus endoderm and ectoderm specification are not implicated in the Yap dependent defects in cell migration, indicating a role of Yap as purely involved in the embryo biomechanical morphogenesis. It additionally and interestingly suggests the existence of a mechanical feedback loop based on Yap mechanosensitivity, robustly orchestrating cell migration with underlying cytoskeleton changes required for cell migration.

We thank this reviewer for the positive comments on our work. We have considered all his/her helpful suggestions and performed new experiments accordingly. We are now confident that all the issues raised have been addressed in the revised version. See individual comments below.

Major comments

1- While the authors nicely show defects in cell migration in Yap double knock-outs, it is not clear whether Yap is the motor and/or the polarizer of cell migration. Indeed, Yap could be more generically required for the expression of cytoskeletal organization and cell-ECM adhesion genes that are necessary for the full mechanical integrity of the multi-cellular tissue with other signaling pathways (downstream of embryo's biochemical patterning?) involved in directed migration. This in such a way migration defects will necessarily appear in Yap knock downs in the presence of the signaling pathways driving cell migration. Indeed, (Porazinski, S. et al. 2015) article (1) (ref 19 of the manuscript) showed that Yap is required for the maintenance of the overall stiffness of medaka embryos to maintain its overall shape.

Therefore, a role of Yap in systematically re-enforcing tissue stiffness by mechanical stimulation of the expression of genes involved in cytoskeletal organization and cell-ECM adhesion in response to morphogenetic movements regulated by other signaling pathways to avoid tissue mechanical alteration in

reponse to forces, that would compromise biomechanical morphogenesis, would be highly interesting as well, and should be tested.

To test this alternative hypothesis, the authors should find a way to apply mechanical forces on the dorsal tissue of the embryo before migration begins, and test in immunofluorescence for the presence, concentration and the location of the major cytoskeleton and cell-ECM adhesion elements downstream of Yap, and for the integrity of the tissue in the wild type and in Yap double knock-outs.

We do agree with the referee in that providing mechanical stimulation experiments will improve the quality of the paper reinforcing our main conclusion on the role of Yap proteins in gastrulation. In the revised version we have used a customized Univert mechanical tester for simultaneous uniaxial compression of 70 % epiboly medaka embryos (see new Figure 9 and S8). We show a significant activation of Yap target genes (i.e. *marcks1b*, *lamc1*, *cyr61*, *ctgfa*) in compressed embryos (20% axial deformation for 20 min), as determined by quantitative FISH and qPCR. These results demonstrate that the Yap-dependent transcriptional program responds to direct mechanical stimulation, indicating that Yap is indeed involved in a mechano-regulatory loop in the context of gastrulation (see also new Figures 5, 8 and S7 that further support this conclusion).

Regarding the possibility of Yap being involved in the mechanical integrity/stiffness of the tissue, we do agree with this referee in that the mechanical properties of the ECM and the adhesion of the cells to the substrate must be altered in the mutants due to the differential transcriptional rate of ECM encoding genes. However, these defects are not sufficient to compromise the survival of the cells, as cell death is normal in the mutants (Figure S1), neither to completely block the migratory capacity of the cells, which are still able to migrate towards the midline, though with lower speed and persistence (Figure 2). We also believe, and so it is acknowledged in the discussion, that “Yap signaling cooperates with other mechanisms to direct cell migration” during gastrulation. The role of Yap in assuring tissue stiffness in response to other signaling pathways is an outstanding question that deserve thorough investigation, but that is out of the scope of this study.

2- The proposal of a Yap mechanical activation feedback is interesting as well, but at that stage not demonstrated into the manuscript, because no control parameter purely mechanical in nature is used in the experiments to test this hypothesis. The existence of a mechanosensitive process necessarily requires at least one experiment involving mechanical control parameters of pure physical nature, to exclude any non-mechanical biochemical side effects due to pharmacological treatments. To definitively test the existence of such mechanical feedback loop, the authors should again find a way to apply a mechanical deformation on the embryo, this time at cell migration stage in the presence of the rock inhibitor, and test if the rescue of a mechanical deformation from rock-inhibitor treated tension defective embryos rescues Yap activation and *marcks1b* expression. As well as if *marcks1b* expression cannot be rescued within these conditions in Yap double knock outs.

This comment is in line with the previous point. We are confident that the mechanical perturbation experiments provided in the revised version (Figures 9 and S8), also supported by the new yolk removal experiments (new Figure 5), are sufficient to demonstrate that Yap transcriptional activity depends on direct mechanical stimulation.

We also performed mechanical perturbation experiments in the presence of the rock inhibitor Rockout. The inhibition of rock severally reduced the expression of Yap targets (both by quantitative FISH and qPCR). However, in Rock-inhibited embryos, the activation of the Yap-dependent transcriptional program does not increase significantly upon mechanical stimulation, indicating that Yap mechanotransduction relies on Rock-dependent intracellular tension (Figure 9 and S8).

*Note on the last experiment suggested: Due to technical limitations, we cannot perform mechanical stimulation experiments in double mutant embryos (this would require determining the embryos' genotype beforehand). Nevertheless, given that yap targets are already downregulated both in *yap* mutants and in Rock-inhibited embryos, and that Rock-inhibited embryos do not respond significantly to the mechanical compression, it is fair to assume that yap mutant embryos treated with Rockout will not respond to the stimulation either.

3- To exclude the existence of uncontrolled off-targets of the pharmacological inhibitor of rock, the authors should duplicate their pharmacological experiments with another inhibitor of tension, like Blebbistatin or ML7.

On the basis of this and other reviewers' comments, we have expanded our pharmacological studies by testing, in addition to Rock-inhibitor, the following drugs: Blebbistatin (myosinII inhibitor), Calyculin A (myosinII activator), and Dasatinib (Src kinases family inhibitor, used to block Yap transcriptional activity in tumoral cells). Quantification of *marcks1b* expression levels and Tead-sensor activity in treated 70% medaka embryos shows that decreasing contractile stress or interfering with Src kinases activity (i.e. Rockout, Blebbistatin or Dasatinib) blocks the Yap-dependent transcriptional program; whereas increasing contractility (i.e. using Calyculin A) results in an increased expression of the Yap targets (Figures 8 and S7). In all cases, drug treatments severely impaired the assembly of the embryo axis (Figure S7). We thank the reviewers for suggesting this informative experiment.

4- In the last paragraph of the results ("Yap senses and activates intracellular tension within a positive feedback loop"), the authors say that they check Yap activation within rock inhibited pharmacological conditions. Unless its direct target *marcks1b* expression is tested, the activation of Yap itself is not tested with the Tead-responsive 4xGTIIC enhancer used from the beginning of the manuscript to monitor Yap activation. The authors should test the Yap activation reporter as well.

As mentioned in the previous point we have added a quantification of the Tead-sensor activity in embryos treated with the different drugs (Rockout, Blebbistatin, Dasatinib and Calyculin A). Consistently, we observed a significant reduction (with Blebbistatin and Dasatinib) or increase (Calyculin A) of the Tead-sensor activity in the treated embryos (Figure 8 and S7). The treatment with Rockout resulted in a reduction of the sensor levels, but this reduction was not statistically significant (Figure 8). Taken all pharmacological experiments together, we could conclude that, similarly to *marcks1b* expression, the activity of the Tead-sensor depends on cell contractility during gastrulation.

Minor comments

1- Depending on major point 1- results, the title may be changed.

Encouraged by our new evidences, we have modified the title: "A Yap-dependent mechano-regulatory loop directs cell migration for embryo axis assembly".

2- The species here studied, medaka, should be mentioned in the abstract.

We had already included the word medaka in the abstract. "Here we show that the double knockout of yap and its paralog yap1b in medaka results in an axis assembly failure."

3- Because Nature Communications is a multi-disciplinary journal with a broad scientific audience, the authors should present medaka embryos morphogenetic movements at gastrulation with the help of a scheme in their introduction.

We thank the reviewer for this suggestion. In the introduction and results sections, we already directed the readers to the excellent reviews on the morphogenetic movements involved in gastrulation in different vertebrate models (Solnica-Krezel and Sepich 2012; Keller 2002; Keller 2005). To complement these references, we are now citing an article describing these movements in detail in the medaka embryo (Hirose et al 2004, Development).

Reviewer #2 (Remarks to the Author):

This manuscript from Sousa-Ortega and colleagues addresses the role of Yap proteins in embryonic axis assembly in medaka embryos. They find that loss of both medaka yap paralogs results in a failure of the posterior body axis to form, not from defects in germ layer formation but rather from reduced migration of lateral cells toward the dorsal midline. Within wild-type embryos, Yap activity is highest in these migrating lateral cells, which adopt a flattened morphology and produce multiple actin-based protrusions to facilitate their migration. The authors report that these cells are more rounded and produce fewer protrusions in Yap double mutants, indicating a possible reduction in cortical cell tension. This is consistent with the observation that inhibition of Rho kinase phenocopies cell morphologies of Yap double mutant embryos. The function of Yap during early development, particularly its role in mechano-sensation, are likely to be of interest to a broad audience. However, while the data presented here identify several associations, further experiments are required to establish causal relationships between tissue mechanics, Yap activity, and embryonic cell migration. For this reason, I recommend the following concerns be addressed prior to publication:

We truly appreciate the reviewer constructive comments of our work. In the revised version we have implemented the suggestions, which have strengthened our conclusions, particularly by providing mechanical stimulation experiments. We are confident that all the issues raised by this referee have been properly addressed.

1. To enhance accessibility by color-blind readers, please switch away from red-green color schemes in figures 3, 5, 6, S4, and if possible 2D-E.

Following the reviewer suggestion and the journal formatting recommendations, we have replaced the corresponding panels so that they are accessible to color-blind readers.

2. Please report sample (n) and trial (N) numbers for all experiments shown.

We have scanned the text and figures to provide this information in the revised version. When needed this information was added (please see Track changes document).

3. Are trajectories and behaviors of midline cells (where Yap is less active) also affected in Yap double mutant embryos? The authors speculate that lateral cells are primarily affected because they exhibit the highest levels of Yap activity, but without examining the behavior of Yap-low midline cells, we can't know that lateral cells are specifically or particularly affected. Further analysis of the midline cell population may provide insights into the function of low-level Yap activity, secondary effects of defective lateral cell migration, and/or cell non-autonomous roles for Yap.

This is a very good suggestion. In the revised version we have measured the migration and persistence parameters also for cells at the midline. In the revised version we show no significant differences in the migratory behavior of wild type and mutant cells at the midline (Figure S2).

4. The comparison of previously reported Yap Dam-ID seq results with differential gene expression analysis in yap mutants in figure 3 is important, but it could be reported in a more informative way. Rather than comparing only the numbers of genes shared between these datasets, I want to know what types of genes are direct Yap targets? Are any GO terms/gene sets enriched among these direct targets?

This question was partially addressed in Supplementary dataset 2, which shows a list of genes associated to the relevant GO terms (e.g. integrin, ECM, cell adhesion or cytoskeleton) indicating which ones are putative direct targets. In the revised version we have also examined enriched GO terms only for the list of genes identified as putative direct targets. The list of GO terms is similar to that obtained by analyzing the complete list of DEGs. This new list however, includes less significant terms as it derives from less genes. The supplementary dataset 2 has been modified to include the new information.

5. Figure 4 shows that the transgenic TEAD reporter is active primarily in lateral cells, while Figure S4A shows GFP within the midline cells but not lateral cells. What explains this discrepancy?

The expression of the Tead reporter is dynamic through development. Its activity is restricted to lateral cells during the convergence and extension movements (stage 14-17: 15 to 22 hpf), the window considered in this article. However, later on at stage 18 (at 26 hpf: once gastrulation has finished and optic buds become apparent at late nerula) the sensor gets activated at the anterior nervous system (Figure S5 now). To make this point clearer, we are now including an explanatory note in this figure legend.

6. The authors identify a correlation between increased cell density and decreased Yap activity. This finding would be more impactful if they could establish a causal relationship. Can one experimentally increase or decrease cell density within medaka embryos to test if it modulates Yap activity?

We thank this reviewer for suggesting this experiment that has been incorporated to the revised version of the work as a main figure. To increase cell density, we extracted yolk material from 50% epiboly embryos using microcapillaries. This approach results in a reduction of the embryo perimeter ($\approx 40\%$) with a significant increase in the inner-mass cell density (Figure 5). In yolk-reduced embryos, we observed a significant reduction of *marcks1b* expression and Tead sensor activity (Figure 5), indicating that increased cell density leads to a downregulation of the Yap-dependent transcriptional program.

7. The authors show that cell morphology, protrusions, and cytoskeletal organization appear different between WT and Yap double mutants, but these data would be stronger if quantified. Number/length of protrusions or number/localization of focal adhesions, for example. It would be easier to quantify the features of individual cells if the membrane marker were expressed in a mosaic fashion.

Following the reviewers' suggestion, we are providing now quantitative data showing significant differences between WT and Yap double mutant embryos for a number of parameters including: cell compactness index, number of filopodial protrusions, total FA length (Fig 6 and 7) and the length of each individual FA (Fig S6C). A detailed description of the quantification methods is also provided in the revised version.

8. The authors show that inhibiting Rho kinase reduces expression of Yap target gene *marcks1b*. Can this decrease in Yap activity also be seen using the live TEAD reporter? Conversely, could increasing myosin contractility using calyculin A increase Yap activity in WT embryos? Could this increased contractility rescue any aspects of the Yap mutant phenotype?

Finally, ROCKOUT phenocopies some aspects of Yap mutants (such as cell flatness). Does ROCKOUT treatment also inhibit embryonic axis assembly similar to loss of Yap?

On the basis of this and other reviewers' comments, we have expanded our pharmacological studies by testing, in addition to Rock-inhibitor, the following drugs: Blebbistatin (myosinII inhibitor), Calyculin A (myosinII activator), and Dasatinib (Src kinases family inhibitor, used to block Yap transcriptional activity in tumoral cells). We have also performed new quantification of *marcks1b* expression levels and of Tead-sensor activity in treated 70% medaka embryos. This shows that decreasing contractile stress or interfering with Src kinases activity (i.e. Rockout, Blebbistatin or Dasatinib) blocks the Yap-dependent transcriptional program; whereas increasing contractility (i.e. using Calyculin A) results in an increased expression of the Yap targets (Figures 8 and S7). In the particular case of Rockout, we could observe a clear reduction of the Tead sensor activity, but that was not statistically significant. However, the reduction of the Tead sensor activity with Blebbistatin and Dasatinib, as well as its increase with CalyculinA, were significant. Thus, we concluded that, similarly to *marcks1b* expression, the activity of the Tead-sensor depends on cell contractility during gastrulation.

In all cases, drug treatments severely impaired the assembly of the embryo axis (Figure S7).

We thank the reviewers for suggesting these complementary experiments to further confirm our previous observations.

Typographical errors:

1. Page 2, near the middle: “being dysregulated mechanical feedbacks a common landmark...”
2. Page 5, top paragraph: “... further analysis on yap double mutants most severe phenotype.”

All typos have been corrected in the revised version (please see Track changes document).

Reviewer #3 (Remarks to the Author):

In this report, Sousa-Ortega et al analysed yap1 and yap1b double mutant embryos that did not properly form the body axis during gastrulation. RNA sequencing, together with the DamID data they previously reported, showed that yap1 and yap1b are required for the expression of genes involved in cytoskeleton reorganization and cell-cell and cell-matrix adhesion. They next showed that Yap was active in cells migrating to the midline. In those cells, the amount of actin and the number of FAs were reduced, and their nuclei were more spherical. Finally, inhibition of Rho/Rock during gastrulation lead to rounder nuclei and a reduction in Yap-mediated transcription. They concluded that Yap activity regulates FAs and actomyosin, and that this mechanism forms a positive feedback loop.

This is a follow-up study of their previous report that provides some useful information. In their previous report, the authors identified Yap and Yap1b binding sites by using the DamID approach. They demonstrated that Yap and Yap1b have transcriptional targets controlling actin and FA formation and claimed that Yap controls mechanical properties of cells during gastrulation and forms a feedback loop. However, it has previously been shown that YAP acts as a mechano-effector from the analysis of the medaka YAP1 mutant by Porazinski et al (Porazinski et al, Nature 2015) and together with YAP's role as a mechano-transducer, they proposed the concept of a mechanical feedback loop. Furthermore, the author's claims about "direct target genes" and "feedback loop" are not fully supported by their results.

We have carefully addressed all the comments raised. In the revised version, we are now providing mechanical stimulation experiments that strengthen our conclusions on Yap controlling a mechano-regulatory loop that directs cell migration during embryo axis assembly. In addition, we have polished our arguments on Yap-dependent transcriptional activation and correct all minor suggestions (Please see Track changes document).

Major points:

1. In their previous report, they identified Yap1 and Yap1b binding genome sites using DamID, and suggested that Yap1 and Yap1b have similar transcriptional targets. In this report, by combining DamID data with RNA-seq data, they identified direct target genes that control the formation of actin and FAs as below. However, this conclusion is not fully supported by the results.

(1) In the abstract, they claim, "Accordingly, we identified genes involved in cytoskeletal organization and cell-ECM adhesion as "direct" Yap targets

(2) In P8, L12, the authors state, "marcks1b expression, a direct target of Yap".

(3) In P9, L28, the authors claim that "a significant proportion of these identified genes are direct target of yap1/yap1b..."

To claim this, the authors must demonstrate, (a) physical binding of Yap1 and Yap1b to the promoter/enhancer of target genes, (b) a reporter assay using the promoter/enhancer-binding sequence, and (c) a reporter assay using the mutated promoter/enhancer-binding sequence.

We disagree with the reviewer's stringent definition of "direct target". For a broad community working on functional genomics, to show binding to a nearby regulatory region in combination with dysregulated gene expression upon mutation of the transcriptional factor (TF) coding gene, would be accepted as standard evidence for direct regulation. This is particularly the case for our dataset, as many of the genes targeted by yap1 and yap1b in medaka were also identified as targeted genes by Chip-seq in previous studies, regardless of the cell type and developmental stage considered (Lian et al 2010; Estaras et al 2017; Zanconato et al 2015; see Figure S12 and table S6 in Vazquez-Marin et al 2019). Regulatory evidence derived from several species/tissues has been found for many key genes in our study, such as *cyr61*, *ctgf*, *marcks1b*, *itgb1*, or *lamc1*.

We concede to this referee that to ultimately show a direct regulation, any given cis-regulatory element should be mutated individually *in vivo* by CRISPR genome editing (i.e. the use of out-of-context reporter assays will be suboptimal for such a rigorous definition of direct targeting), something that is out of the scope of this work. To make this point more clear in the revised version, we have decided to tone down our claim

substituting the term “*direct target*” for “*putative/potential direct target*” and including a precise definition of the term: i.e. genes for which Yap binding to nearby regulatory regions has been identified by DamID-seq and that appear significantly downregulated in yap double mutants”. In any case, and beyond the definition of “direct targeting”, our conclusions are based on whole-genome transcriptomics and chromatin binding assays rather than on the individual behavior of a given element/gene, and thus we think they remain valid.

2. In the section titled “Yap senses and activates intracellular tension...”, the authors claim to show that Yap activity regulates FAs and the actomyosin cytoskeleton, which allows for the coupling of intracellular tension to the ECM and suggest the existence of mechano-regulatory feedback. Porazinski et al. previously reported the Yap’s newly discovered function as a mechano-effector in which medaka YAP mutants showed reduction of tissue tension and aberrant ECM formation. They also suggested the presence of a mechano-regulatory feedback together with Yap’s previously reported mechano-transducer functions. Furthermore, the author’s description that “Yap senses...” is not correct. Yap is a mechano-transducer, but it is not a mechano-sensor (Dupont et al., Nature 2011).

The work by Porazinski and coworkers describing the phenotype of *yap1* single mutants in medaka, is certainly a reference in the field as it highlighted the important role of Yap proteins in maintaining tissue morphogenesis and intracellular tension *in vivo*. We have referred to this article several times in the text. However, there are fundamental differences between the mechanical feedback loop suggested by Porazinski et al and that described in our work. Whereas Porazinski et al reported an *arhgap18*-dependent **negative** feedback loop involved in organogenesis (particularly during optic cup development), the Yap-dependent mechano-regulatory loop here described is a self-sustained positive loop active in migratory cells converging towards the midline. It is important to emphasize that our work about the role of Yap proteins focus on an earlier gastrulating stage that was not described previously in the work of Porazinski et al., since in order to observe a clear phenotypic difference at this stage is required to analyze *yap1/yap1b* double mutants and the work of Porazinski et al. is limited to *yap1* single mutants.

Regarding the definition of Yap as a mechano-transducer or a mechano-sensor, we think that both are accepted by the scientific community. Even in the original paper by Dupont et al. quoted by the reviewer, one of the main sections is titled “Yap/Taz sense cytoskeletal tension”. Additional examples of Yap being acknowledged as a mechano-sensing protein can be found in very influential papers and reviews (Low et al 2014 FEBS Letter; Nardone et al 2017 Nat Comm; Lin et al 2017 eLife, Elosegui-Artola 2017 Cell). In agreement with this, in our work we are referring to Yap proteins both as mechano-transducers and mechano-sensors.

3. The authors drew their conclusion on YAP’s control of mechanical properties based on their inference from the nuclear shape and the actin reporter Utrophin::GFP (Fig. 5, 6A, B). However, nuclear shape does not always correlate with cell compression, and actin over-polymerization leads to lower tissue tension (Pinto et al., Dev Cell, 2012). To make such a statement, they must carry out direct measurements, e.g. laser cutting of actin filaments and micro-pipette analysis.

To support our claim of Yap proteins being involved in a mechano-regulatory loop we are providing mechanical stimulation experiments in the revised version of the work. These experiments, using a customized Univert mechanical tester for embryo compression show that Yap-dependent transcriptional program responds to mechanical tension (see new Figures 9 and S8). The converse regulatory arrow (i.e from Yap activation to intracellular tension) was inferred in our work from nuclear morphology, actin cortical recruitment, and focal adhesions assembly in mutant tissues; and it is now further supported by new measurements of total FA length, cell compactness index, and filopodial protrusions number (Figures 6, 7 and S6). Direct measurements showing a significantly reduced tissue tension have been already performed both in medaka *yap1* mutants and zebrafish gastrulating embryos, using micropipette aspiration and laser ablation experiments respectively (Porazinski. et al 2015). We are now mentioning these experiments that support our conclusions.

“From these measurements, we hypothesized that the noticeable reduction of cortical filamentous actin and FAs observed in yap mutant cells may lead to a decrease in intracellular tension, which is reflected in a more relaxed and rounded cell morphology. This is in agreement with previous observations indicating a reduced

tissue tension in medaka mutants for yap1 and yap/taz knockdown embryos in zebrafish (Porazinski. et al 2015)”.

4. While in the anterior axis, GTIIC:GFP seems not to be active (Fig.4 A), *marcks11b* is expressed (Fig.4 B) and depicted as the YAP+ in D (Fig.4 D). These descriptions appear to be consistent.

The expression of the GTIIC:GFP sensor is dynamic through development. Its activity is restricted to lateral cells during the convergence and extension movements (stage 14-17: 15 to 22 hpf), the window considered in this article. However, later on at stage 18 (at 26 hpf: once gastrulation has finished and optic buds become apparent at late nerula) the sensor gets activated at the anterior axis (Figure S5 now).

This is consistent with the expression of *marcks11b* in the anterior axis shown in Figure 4. The inconsistency might be that the expression of the GTIIC:GFP Tead sensor in the anterior axis is a bit delayed during development. This is just explained by the different nature of looking at Yap activation by these two approaches; while the changes in *marcks11b* expression will be detected immediately by *in situ* hybridization techniques, the activation of the GTIIC:GFP Tead sensor will require the additional time of the GFP protein translation and folding (at least one additional hour, but it varies depending on the biological context).

Minor points:

1. P2 L8: ESC (Embryonic Stem Cell)
2. P2 L22: well-known transcriptional regulators “that are” able to....
3. P3 L18: role for the mechanosensory Yap, mechano-transducer
4. P3 L36: “without an apparent definition”, “formation” is better.
5. P4 L11: we asked ourselves if...
6. P5 L5: the mechanism behind Yap, “underlying” is better
7. P7 L2: Pax::mKate mRNA, needs a reference.
8. P9. L21 However, it is very unlikely... This sentence is unclear.
9. P21. Ref 20 and 23 are lacking information about the number of pages.

All these minor changes have been corrected in the revised version (please see Track changes document).

REVIEWER COMMENTS

Reviewer #1 (Remarks to the Author):

The revised manuscript "A Yap-dependent mechano-regulatory loop directs cell migration for embryo axis assembly" by Ana Sousa-Ortega, Javier Vazquez-Marin, Juan R. Martinez-Morales and colleagues nicely produced a substantial amount of new successful experiments to answer the reviewer's questions.

The use of other drugs, Blebbistatin, Calyculin A, and Dasatinib, to exclude the role of any secondary target of the Rock inhibitor, consistently showed that tension can regulate Yap activity and downstream gene expression like *marcksl1b* (point 3). The authors also added the Yap activity sensor test when absent from the experiment (point 4). The authors additionally addressed minor points.

- Regarding point 1, the authors found that the expression of Yap target genes can be directly mechanically stimulated before the migration stage, which is consistent with the fact that the Yap pathway is mechanosensitive in medaka embryos at 70% epiboly. However, the manuscript does not specify whether control (non-compressed) embryos are confined in the compression chamber with the position of the indenter stopped just before compression. Indeed, the confinement itself could lead to a response of Yap-dependent genes to hypoxia. If not, it would be important to check this point.

- Regarding point 2, the fact that mechanical deformation does not rescue Yap-dependent transcriptional programs in Rock-treated defective embryos can be interpreted either by following the authors' suggestion of the need for Rock-dependent tension to induce mechanical stimulation of the Yap pathway, or by the fact that mechanical rescue of tension in defective embryos cannot rescue Yap pathway activation at the migration stage. The authors should thus be more cautious in saying that they have shown the existence of such a loop in the discussion, as other possibilities are still open.

Since such a rescue experiment from a mechanical defective state is important to definitively support the concept of a mechanoregulatory loop, the authors could try the experiments with Blebbistatin.

- Finally, since the authors also cannot rule out that their observations are based on a more general role of Yap in maintaining tissue integrity, they could also be more careful in the title, replacing "direct" with "required" for example, and more explicitly discuss this possibility based on ref 19.

Reviewer #2 (Remarks to the Author):

The authors have addressed all of my concerns in this revised manuscript. The methods for directly manipulating mechanical forces and the expanded set of pharmacological perturbations in particular provide compelling evidence in support of the proposed mechano-regulatory loop.

Reviewer #3 (Remarks to the Author):

Please see attached word document.

Manuscript: NCOMMS-22-22583A: A Yap-dependent mechano-regulatory loop directs cell migration for embryo axis assembly: Ana Sousa-Ortega ^{1*}, Javier Vázquez-Marín ^{1*}, Estefanía Sanabria-Reinoso ¹, Jorge Corbacho¹, Rocío Polvillo ¹, Alejandro Campoy-López ¹, Lorena Buono ¹, Felix Loosli ², María Almuedo-Castillo ^{1#}, Juan R. Martínez-Morales ^{1#}.

REVIEWER COMMENTS (Point by Point answer)

Reviewer #3 (Remarks to the Author):

In this report, Sousa-Ortega et al analysed yap1 and yap1b double mutant embryos that did not properly form the body axis during gastrulation. RNA sequencing, together with the DamID data they previously reported, showed that yap1 and yap1b are required for the expression of genes involved in cytoskeleton reorganization and cell-cell and cell-matrix adhesion. They next showed that Yap was active in cells migrating to the midline. In those cells, the amount of actin and the number of FAs were reduced, and their nuclei were more spherical. Finally, inhibition of Rho/Rock during gastrulation lead to rounder nuclei and a reduction in Yap-mediated transcription. They concluded that Yap activity regulates FAs and actomyosin, and that this mechanism forms a positive feedback loop.

This is a follow-up study of their previous report that provides some useful information. In their previous report, the authors identified Yap and Yap1b binding sites by using the DamID approach. They demonstrated that Yap and Yap1b have transcriptional targets controlling actin and FA formation and claimed that Yap controls mechanical properties of cells during gastrulation and forms a feedback loop. However, it has previously been shown that YAP acts as a mechano-effector from the analysis of the medaka YAP1 mutant by Porazinski et al (Porazinski et al, Nature 2015) and together with YAP's role as a mechano-transducer, they proposed the concept of a mechanical feedback loop. Furthermore, the author's claims about "direct target genes" and "feedback loop" are not fully supported by their results.

We have carefully addressed all the comments raised. In the revised version, we are now providing mechanical stimulation experiments that strengthen our conclusions on Yap controlling a mechano-regulatory loop that directs cell migration during embryo axis assembly. In addition, we have polished our arguments on Yap-dependent transcriptional activation and correct all minor suggestions (Please see Track changes document).

The authors addressed the major points 1 and 4, but did not adequately address 2 and 3. They did not deal with this reviewer's concerns about their proposed mechanical model and mechanical property measurements. Therefore, the added data in this revision do not fully support the positive feedback "mechano-regulatory loop" model drawn by the authors.

Major points:

2. In the section titled "Yap senses and activates intracellular tension...", the authors claim to show that Yap activity regulates FAs and the actomyosin cytoskeleton, which allows for the coupling of intracellular tension to the ECM and suggest the existence of mechano-regulatory feedback. Porazinski et al. previously reported the Yap's newly discovered function as a mechano-effector in which medaka YAP mutants showed reduction of tissue tension and aberrant ECM formation. They also suggested the presence of a mechano-regulatory feedback together with Yap's previously reported mechano-transducer functions. Furthermore, the author's description that "Yap senses..." is not correct. Yap is a mechano-transducer, but it is not a mechano-sensor (Dupont et al., Nature 2011).

The work by Porazinski and coworkers describing the phenotype of yap1 single mutants in medaka, is certainly a reference in the field as it highlighted the important role of Yap proteins in maintaining tissue

morphogenesis and intracellular tension *in vivo*. We have referred to this article several times in the text. However, there are fundamental differences between the mechanical feedback loop suggested by Porazinski et al and that described in our work. Whereas Porazinski et al reported an *arhgap18*-dependent **negative** feedback loop involved in organogenesis (particularly during optic cup development), the Yap-dependent mechano-regulatory loop here described is a self-sustained positive loop active in migratory cells converging towards the midline. It is important to emphasize that our work about the role of Yap proteins focus on an earlier gastrulating stage **that was not described previously in the work of Porazinski et al.**, since in order to observe a clear phenotypic difference at this stage is required to analyze *yap1/yap1b* double mutants and the work of Porazinski et al. is limited to *yap1* single mutants.

The authors falsely claim that an earlier gastrulating stage phenotype was not described previously in the work of Porazinski et al. Indeed, Porazinski et al., previously reported in Fig.1b and in quantitative analysis in c that the medaka YAP1 mutants exhibited delayed blastopore closure and reduced F-actin in the envelope layer (EVL) during gastrulation. They also showed that zebrafish YAP and TAZ double knockdown embryos exhibited delayed blastopore closure and even reduction of actomyosin ring contraction by mechanical measurement using laser cutting. Therefore, the author’s findings of the early gastrulating phenotype in YAP mutants are limited.

In the revised version, the authors placed even more stress on the “**mechano-regulatory loop**” by changing the title to: “A Yap-dependent mechano-regulatory loop directs cell migration for embryo axis assembly”. In the abstract, “Our results **indicate** that Yap coordinates a **mechano-regulatory loop** to sustain intracellular tension and maintain the directed cell migration for embryo axis development”.

Unfortunately, these conclusions on the mechano-regulatory loop were not fully supported because of the lack of data showing that the YAP target gene mediates a mechano-regulatory “loop”. There are many other potential mechanisms by which YAP activation could lead to actomyosin contraction.

(1) Firstly, the authors did not clarify what are the components of the “mechano-regulatory loop” in the Results section. This reviewer assumes that the loop could be ① mechanical stimuli > ② YAP activation > ③ *marcks1b* transcription > ④ (data linking ③ & ⑤ missing) > ⑤ F-actin accumulation > ⑥ actomyosin contraction (cell mechanical properties not measured) > ① as shown in this reviewer’s figure below. To clarify this, they have to describe components of the feedback loop in the Results section.

- (2) While the authors carried out the following 3 experiments, these experiments do not fully support their model of mechano-regulatory positive feedback because of the lack of evidence that YAP and actomyosin is linked by the target gene as shown ④ in this reviewer's figure.

The 3 experiments were to examine whether mechanical positive feedback is operating in migrating cells, as shown in (B) this reviewer's figure. Modulation of mechanical properties by (a) reduction of cortical tension by inhibiting actomyosin contraction using Rockout (blocking ⑥-①), (b) increased F-actin polymerization by CalyculinA (activating ⑥-①), and (c) mechanical compression (①), inactivated YAP in (a) or activated YAP in (b, c), and reduction or induction of ③ *marcks1b* expression.

To fill up the gap, the authors have to show that *marcks1b* overexpression rescues the YAP double mutant phenotype.

- (3) What is the molecular mechanism that links *marcks1b* and actin localization or actomyosin contraction? They cited the ref. 34, but the molecular mechanism by which *marcks1b* controls actomyosin contraction is not well understood. Thus, the authors have to show the mechanism, e.g. as shown in ref.19 that ARHGAP18 suppresses F-actin polymerization to optimize actomyosin contraction.
- (4) P6 L23. While the authors showed that *marcks1b* expression coincided with YAP activation in lateral cells and its expression was reduced in YAP double mutants, it remains unclear whether *marcks1b* is indeed required for the cortical actin phenotype and cell migrations, and also for activating YAP if **positive feedback** works.

To distinguish (a) compression activate YAP via **positive** feedback mediated by *marcks1b* from (b) alterations of cell mechanical properties activate YAP and *marcks1b* as reported in many papers, the authors have to show that *marcks1b* overexpression could increase actomyosin contraction and YAP activation in WT embryos and conversely, knock-down of *marcks1b* leads to a reduction of cortical actin & cell tension, and YAP inactivation.

3. The authors drew their conclusion on YAP's control of mechanical properties based on their inference from the nuclear shape and the actin reporter Utrophin::GFP (Fig. 5, 6A, B). However, nuclear shape does not always correlate with cell compression, and actin over-polymerization leads to lower tissue tension (Pinto et al., Dev Cell, 2012). To make such a statement, they must carry out direct measurements, e.g. laser cutting of actin filaments and micro-pipette analysis.

To support our claim of Yap proteins being involved in a mechano-regulatory loop we are providing mechanical stimulation experiments in the revised version of the work. These experiments, using a customized Univert mechanical tester for embryo compression show that Yap-dependent transcriptional program responds to mechanical tension (see new Figures 9 and S8). The converse regulatory arrow (i.e from Yap activation to intracellular tension) was inferred in our work from nuclear morphology, actin cortical recruitment, and focal adhesions assembly in mutant tissues; and it is now further supported by new measurements of total FA length, cell compactness index, and filopodial protrusions number (Figures 6, 7 and S6). Direct measurements showing a significantly reduced tissue tension have been already performed both in medaka *yap1* mutants and zebrafish gastrulating embryos, using micropipette aspiration and laser ablation experiments respectively (Porazinski. et al 2015). We are now mentioning these experiments that support our conclusions.

"From these measurements, we hypothesized that the noticeable reduction of cortical filamentous actin and FAs observed in yap mutant cells may lead to a decrease in intracellular tension, which is reflected in a more

relaxed and rounded cell morphology. This is in agreement with previous observations indicating a reduced tissue tension in medaka mutants for yap1 and yap/taz knockdown embryos in zebrafish (Porazinski. et al 2015)”.

The authors inferred mechanical properties via F-actin, FA accumulation and nuclear shape. However, this inference is misleading. For example, F-actin accumulation and its cell cortex tension do not correlate linearly. As F-actin accumulates, the cell tension increases then decreases (Chugh *et al.*, Nat Cell Biol 2015). Although nuclear deformation is required for YAP nuclear localization, deformed nuclei do not necessarily have nuclear YAP. Thus, nuclear deformation can not be used for inferring tissue mechanics.

Thus, direct measurement of the mechanical properties of migrating cells is definitely required to support their conclusion and model. While this reviewer suggested direct measurement of the mechanical properties of cells in the previous review, but the authors did not deal with this suggestion. This reviewer stresses that they have to show alteration of mechanical properties by direct measurements e.g. micropipet aspiration done in tissues (ref.19) and in single cells of zebrafish e.g. Kardash *et al.*, Nat Cell Biol 2010.

Although Porazinski *et al.*, demonstrated reduced TISSUE tension of the blastomere ring and the neural tube, they did not measure CELL tension in migrating cells during gastrulation. It is possible in fish embryos by micropipette aspiration as shown previously (e.g. Kardash *et al.*, Nat Cell Biol 2010).

Minor points

1. In the hippo-YAP field, we use YAP rather than Yap e.g. in the review by Guan. (Ann Rev Biochem, 2019)
2. P2. L15, P3. L9 “acto-myosin” should be actomyosin.
3. P9 L29, Spell out WNT-PC should be spelt out.

**REVIEWER COMMENTS (Point by point answer). Manuscript: NCOMMS-22-22583A:
A Yap-dependent mechano-regulatory loop directs cell migration for embryo axis assembly:**

Reviewer #1 (Remarks to the Author):

The revised manuscript "A Yap-dependent mechano-regulatory loop directs cell migration for embryo axis assembly" by Ana Sousa-Ortega, Javier Vazquez-Marin, Juan R. Martinez-Morales and colleagues nicely produced a substantial amount of new successful experiments to answer the reviewer's questions.

The use of other drugs, Blebbistatin, Calyculin A, and Dasatinib, to exclude the role of any secondary target of the Rock inhibitor, consistently showed that tension can regulate Yap activity and downstream gene expression like *marcksl1b* (point 3). The authors also added the Yap activity sensor test when absent from the experiment (point 4). The authors additionally addressed minor points.

We thank the reviewer for the positive comments on the revised version, as well as the previous suggestions that helped us to improve substantially our work. Please, find here a point by point answer to the minor issues still pending.

- Regarding point 1, the authors found that the expression of Yap target genes can be directly mechanically stimulated before the migration stage, which is consistent with the fact that the Yap pathway is mechanosensitive in medaka embryos at 70% epiboly. However, the manuscript does not specify whether control (non-compressed) embryos are confined in the compression chamber with the position of the indenter stopped just before compression. Indeed, the confinement itself could lead to a response of Yap-dependent genes to hypoxia. If not, it would be important to check this point.

In our experiments non-compressed embryos were confined in parallel in an independent mock chamber. **We are now including this information in the methods section to make this point clear.** In any case a potential embryo response to hypoxia is quite unlikely in our case. The compression chamber, filled with 9 ml of E3 medium, is not a sealed compartment. On the contrary, when the indenter piece moves down axially, the medium but not the embryos is freely displaced at the edges (i.e. a 100 μm gap separates the indenter from the walls of the chamber). This medium is in free contact with the air allowing its oxygenation. In addition, our experiments entailed a short compression time (20 min), whereas sustained exposure to hypoxic conditions (most probably hours) will be required to trigger a hypoxia response in the medaka embryos (see Mu et al 2017 *Aquat Toxicol*).

- Regarding point 2, the fact that mechanical deformation does not rescue Yap-dependent transcriptional programs in Rock-treated defective embryos can be interpreted either by following the authors' suggestion of the need for Rock-dependent tension to induce mechanical stimulation of the Yap pathway, or by the fact that mechanical rescue of tension in defective embryos cannot rescue Yap pathway activation at the migration stage. The authors should thus be more cautious in saying that they have shown the existence of such a loop in the discussion, as other possibilities are still open.

Since such a rescue experiment from a mechanical defective state is important to definitively support the concept of a mechanoregulatory loop, the authors could try the experiments with Blebbistatin.

As the reviewer pointed out in the introductory paragraph, in our revised manuscript we included experiments performed with different types of drugs, including Rockout, Blebbistatin, Calyculin A, and Dasatinib. This allow us to exclude the involvement of any secondary target of the Rock inhibitor (Rockout), and to be more confident about the idea that Yap activation depends on changes in the intracellular tension; as demonstrated in many independent studies. Nevertheless, to be more cautious with the interpretation of the compression experiment summarized in figure S8 (S9 in the last revised version), **we have changed the wording of the following statement:** “Finally, we checked the effect of mechanical compression in embryos after pharmacological inhibition by Rockout treatment. We saw that, in the absence of Rock-mediated cytoskeletal tension, the impact of the mechanical perturbation on the yap-dependent transcriptional activation is largely reduced (Fig S9), indicating that Yap ability to mechanotransduce relies on the generation of intracellular tension.”

In the revised version being: “Finally, we checked the effect of mechanical compression in embryos after pharmacological inhibition by Rockout treatment. We saw that, in Rock inhibited embryos, the impact of the mechanical perturbation on the yap-dependent transcriptional activation is largely reduced (Fig S9). This observation is in agreement with our previous observations (Fig 8), and most probably indicates that Yap ability to mechanotransduce relies on the generation of intracellular tension.”

- Finally, since the authors also cannot rule out that their observations are based on a more general role of Yap in maintaining tissue integrity, they could also be more careful in the title, replacing "direct" with "required" for example, and more explicitly discuss this possibility base on ref 19.

As an alternative title we suggest: **"A Yap-dependent mechano-regulatory loop is essential during convergent cell migration for embryo axis assembly"**. We leave to the Editor the last decision on this change. To discuss the possibility of a general role of Yap in tissue integrity, the following sentence has been introduced in the discussion:

“A general role of Yap in maintaining tissue integrity and shape during organogenesis has been postulated¹⁹. Here, our results suggest that, in the context of gastrulation, Yap plays a pivotal role in the establishment of a mechano-regulatory feedback loop (Fig 9D).”

Reviewer #2 (Remarks to the Author):

The authors have addressed all of my concerns in this revised manuscript. The methods for directly manipulating mechanical forces and the expanded set of pharmacological perturbations in particular provide compelling evidence in support of the proposed mechano-regulatory loop.

We are glad to read that this reviewer agrees with our conclusions on the existence of a mechano-regulatory loop directing axis formation. We thank the reviewer for her/his critical and constructive input, which has helped us significantly to strengthen the main conclusions of our work.

Reviewer #3 (Remarks to the Author):

In this report, Sousa-Ortega et al analysed yap1 and yap1b double mutant embryos that did not properly form the body axis during gastrulation. RNA sequencing, together with the DamID data they previously reported, showed that yap1 and yap1b are required for the expression of genes involved in cytoskeleton reorganization and cell-cell and cell-matrix adhesion. They next showed that Yap was active in cells migrating to the midline. In those cells, the amount of actin and the number of FAs were reduced, and their nuclei were more spherical. Finally, inhibition of Rho/Rock during gastrulation lead to rounder nuclei and a reduction in Yap-mediated transcription. They concluded that Yap activity regulates FAs and actomyosin, and that this mechanism forms a positive feedback loop.

This is a follow-up study of their previous report that provides some useful information. In their previous report, the authors identified Yap and Yap1b binding sites by using the DamID approach. They demonstrated that Yap and Yap1b have transcriptional targets controlling actin and FA formation and claimed that Yap controls mechanical properties of cells during gastrulation and forms a feedback loop. However, it has previously been shown that YAP acts as a mechano-effector from the analysis of the medaka YAP1 mutant by Porazinski et al (Porazinski et al, Nature 2015) and together with YAP's role as a mechano-transducer, they proposed the concept of a mechanical feedback loop. Furthermore, the author's claims about "direct target genes" and "feedback loop" are not fully supported by their results.

R1: We have carefully addressed all the comments raised. In the revised version, we are now providing mechanical stimulation experiments that strengthen our conclusions on Yap controlling a mechano-regulatory loop that directs cell migration during embryo axis assembly. In addition, we have polished our arguments on Yap-dependent transcriptional activation and correct all minor suggestions (Please see Track changes document).

The authors addressed the major points 1 and 4, but did not adequately address 2 and 3. They did not deal with this reviewer's concerns about their proposed mechanical model and mechanical property measurements. Therefore, the added data in this revision do not fully support the positive feedback "mechano-regulatory loop" model drawn by the authors.

R2: In the revised version we have made an effort to address all the remaining concerns, either by providing additional experiments or by clarifying our arguments in the text and the reply letter (please see also the Track changes document).

Major points:

2. In the section titled “Yap senses and activates intracellular tension...”, the authors claim to show that Yap activity regulates FAs and the actomyosin cytoskeleton, which allows for the coupling of intracellular tension to the ECM and suggest the existence of mechano-regulatory feedback. Porazinski et al. previously reported the Yap’s newly discovered function as a mechano-effector in which medaka YAP mutants showed reduction of tissue tension and aberrant ECM formation. They also suggested the presence of a mechano-regulatory feedback together with Yap’s previously reported mechano-transducer functions. Furthermore, the author’s description that “Yap senses...” is not correct. Yap is a mechano-transducer, but it is not a mechano-sensor (Dupont et al., Nature 2011).

R1: The work by Porazinski and coworkers describing the phenotype of *yap1* single mutants in medaka, is certainly a reference in the field as it highlighted the important role of Yap proteins in maintaining tissue morphogenesis and intracellular tension in vivo. We have referred to this article several times in the text. However, there are fundamental differences between the mechanical feedback loop suggested by Porazinski et al and that described in our work. Whereas Porazinski et al reported an *arhgap18*-dependent negative feedback loop involved in organogenesis (particularly during optic cup development), the Yap- dependent mechano-regulatory loop here described is a self-sustained positive loop active in migratory cells converging towards the midline. It is important to emphasize that our work about the role of Yap proteins focus on an earlier gastrulating stage that was not described previously in the work of Porazinski et al., since in order to observe a clear phenotypic difference at this stage is required to analyze *yap1/yap1b* double mutants and the work of Porazinski et al. is limited to *yap1* single mutants.

The authors falsely claim that an earlier gastrulating stage phenotype was not described previously in the work of Porazinski et al. Indeed, Porazinski et al., previously reported in Fig.1b and in quantitative analysis in c that the medaka YAP1 mutants exhibited delayed blastopore closure and reduced F-actin in the envelope layer (EVL) during gastrulation. They also showed that zebrafish YAP and TAZ double knockdown embryos exhibited delayed blastopore closure and even reduction of actomyosin ring contraction by mechanical measurement using laser cutting. Therefore, the author’s findings of the early gastrulating phenotype in YAP mutants are limited.

R2: The main defects observed in our double *yap1;yap1b* medaka mutants are the lack of directed cell migration and axis assembly, and these defects are not observed in single *yap1* medaka mutants. That is why we claim that we uncover new functions of Yap paralogs during medaka gastrulation. Despite the very valued descriptions of the Yap single mutant defects in Porazinski et al., we believe our findings are novel, since we described a new role shared by both *yap* paralogs in the embryo’s primary axis assembly, which is key for the construction of the vertebrate body plan. Moreover, we also described a mechanosensitive mechanism for the convergent migration of the precursor cells to the midline.

In the revised version, the authors placed even more stress on the “**mechano-regulatory loop**” by

changing the title to: “A Yap-dependent mechano-regulatory loop directs cell migration for embryo axis assembly”. In the abstract, “Our results **indicate** that Yap coordinates a **mechano-regulatory loop** to sustain intracellular tension and maintain the directed cell migration for embryo axis development”.

Unfortunately, these conclusions on the mechano-regulatory loop were not fully supported because of the lack of data showing that the YAP target gene mediates a mechano-regulatory “loop”. There are many other potential mechanisms by which YAP activation could lead to actomyosin contraction.

(1) Firstly, the authors did not clarify what are the components of the “mechano-regulatory loop” in the Results section. This reviewer assumes that the loop could be ① mechanical stimuli> ② YAP activation> ③ *marcks1b* transcription> ④ (data linking ③ & ⑤ missing) > ⑤ F-actin accumulation>⑥ actomyosin contraction (cell mechanical properties not measured) >① as shown in this reviewer’s figure below. To clarify this, they have to describe components of the feedback loop in the Results section.

(2) While the authors carried out the following 3 experiments, these experiments do not fully support their model of mechano-regulatory positive feedback because of the lack of evidence that YAP and actomyosin is linked by the target gene as shown ④ in this reviewer’s figure.

The 3 experiments were to examine whether mechanical positive feedback is operating in migrating cells, as shown in (B) this reviewer’s figure. Modulation of mechanical properties by (a) reduction of cortical tension by inhibiting actomyosin contraction using Rockout (blocking ⑥-①), (b) increased F-actin polymerization by CalyculinA (activating ⑥-①), and (c) mechanical compression (①), inactivated YAP in (a) or activated YAP in (b, c), and reduction or induction of ③ *marcks1b* expression. To fill up the gap, the authors have to show that *marcks1b* overexpression rescues the YAP double mutant phenotype.

(3) What is the molecular mechanism that links *marcks1b* and actin localization or actomyosin contraction? They cited the ref. 34, but the molecular mechanism by which *marcks1b* controls actomyosin contraction is not well understood. Thus, the authors have to show the mechanism, e.g. as shown in ref.19 that ARHGAP18 suppresses F-actin polymerization to optimize actomyosin contraction.

(4) P6 L23. While the authors showed that *marcks1b* expression coincided with YAP activation

in lateral cells and its expression was reduced in YAP double mutants, it remains unclear whether *marcks11b* is indeed required for the cortical actin phenotype and cell migrations, and also for activating YAP if **positive feedback** works.

To distinguish (a) compression activate YAP via **positive** feedback mediated by *marcks11b* from (b) alterations of cell mechanical properties activate YAP and *marcks11b* as reported in many papers, the authors have to show that *marcks11b* overexpression could increase actomyosin contraction and YAP activation in WT embryos and conversely, knock-down of *marcks11b* leads to a reduction of cortical actin & cell tension, and YAP inactivation.

R2: The referee is depicting an oversimplified and lineal model of Yap mechanism of action. This model does not correspond to our view, as according to our data *macks11b* is not the main/single effector of Yap. We follow the expression of this gene only as a readout of Yap activity, in addition to our Yap reporter *4xGTIIC::GFP*. In fact, a key aspect of our work is precisely the description of an entire Yap-dependent transcriptional program that includes the **parallel recruitment** of hundreds of genes involved in ECM assembly, cytoskeletal organization, integrin binding, and focal adhesion (see **Fig. 1 for the reviewers, Fig. 3, Table S1, and Supplementary dataset 1**). This parallel recruitment draws a picture that is far from the lineal pathway outlined in the referee scheme.

Figure 1 for the reviewers: Main components of the Yap-dependent transcriptional program encode for proteins that provide a link between the ECM and the actin cytoskeleton.

In our view, it would be unlikely to expect a full rescue of the *yap* double mutant phenotype by *marcks11b* overexpression, or its phenocopy by *marcks11b* knockdown, as many other downstream regulators of the actin cytoskeleton act in parallel. Nevertheless, in order to explore this possibility, we decided to perform a loss-of-function approach, which is less prompt to out-of-context artifacts than overexpression experiments. We took advantage of the CRISPR/Cas13d

that allows the specific and efficient targeting of mRNAs, and has been proved as an efficient tool in teleost models, including medaka (Kushawah et al 2020, Dev Cell). Knockdown of *marcks11b* using Cas13d together with specific gRNAs resulted in an efficient depletion of its mRNA. However, injected embryos displayed only a very mild phenotype, including delayed development and epiboly in 26% of the injected embryos. In all cases, the primary axis was correctly form in *marcks11b* depleted embryos (**Fig2 for the reviewers**). We conclude from this experiment that the Yap-dependent transcriptional program plays a broader role than merely the activation of *marcks11b* to direct axial formation.

Figure 2 for the reviewers: (A) Stacked barplots showing percentage of epiboly in embryos no injected, injected with Cas13d protein alone, or together with *marcks11b* gRNAs. (Nembryos ≥ 30). (B) RT-qPCR analysis of mRNA levels of *marcks11b* in embryos injected only with Cas13d protein and in affected embryos injected with Cas13d and *marcks11b* gRNAs. Data are represented as mean \pm SD; (Nembryos ≥ 8). The housekeeping gene *ef1a* mRNA was used as normalization control. (C) Brightfield images of stage 18 embryos injected only with Cas13d protein and affected embryos injected with Cas13d and *marcks11b* gRNAs. Frontal stereo microscope images of embryos are shown.

Going back to the reviewer scheme, **we strongly believe that the missing connection between Yap activation (2) and cortical actin accumulation (5) is simply the integrin-dependent assembly of focal adhesions (FAs)**. The link between integrin-mediated cell adhesion to the ECM and F-actin accumulation has been demonstrated extensively in the literature and can be considered textbook knowledge: see for example the classical work by Hynes and Destree, 1978 in *Cell*; and the reviews by Hynes *Cell* 1992; Scharz and Gardel 2012 *J. Cell Science*; Bachir et al 2017 *Cold Spring Harb Perspect Biol*; Martino et al. *Front in Physiol* 2018. In our work we showed that FAs, preferentially

oriented towards the YSL in dorsally converging cells, are largely reduced in the double mutants (**Fig 6**). As the link between FAs and cortical actin accumulation is well-established, we reasoned that the reduced cortical F-actin, cell spreading and filopodia observed in the double mutants (**Fig6 and S6**) resulted from the deficient assembly of FAs. To prove this hypothesis, we have performed an additional experiment (now **Fig S7**, see attached) aimed to “rescue” the missing FAs in the double mutant embryos. As restoring all FAs components in the proper stoichiometric ratio is an experiment beyond our capacity, we decided to overexpress the FA component *paxillin*, a central node in the ECM-FA-Actin cytoskeleton axis. Overexpression of *paxillin::mKate* mRNA at 300 pg per embryo (i.e. a higher concentration than in our previous analytical experiments), was now sufficient to partially rescue the mutant defects at the cellular level: including spread cell morphology, cell compactness, filopodia number and nuclear morphology (**Fig S7**).

Figure S7: Paxillin overexpression rescues cell and nuclear morphology in *yap* mutant embryos. **(A)** Confocal microscopy images of dorsally converging cells from WT, *yap1*^{-/-};*yap1b*^{-/-} and Paxillin-rescued *yap1*^{-/-};*yap1b*^{-/-} embryos injected with *Utrophin::GFP* and stained with DAPI. Schematic representation of the embryo indicating the area of interest with a yellow rectangle is shown in the upper left side of the panel. Cell shapes are represented with white lines in the images corresponding to DAPI. **(B)** Quantification of average cell area in WT, *yap1*^{-/-};*yap1b*^{-/-} and Paxillin-rescued *yap1*^{-/-};*yap1b*^{-/-} embryos. **(C)** Quantification of average cell compactness, as determined by the ratio between the cell area and the area of the circle having the same perimeter, in WT, *yap1*^{-/-};*yap1b*^{-/-} and Paxillin-rescued *yap1*^{-/-};*yap1b*^{-/-} embryos. **(D)** Quantification of the average number of filopodia in WT, *yap1*^{-/-};*yap1b*^{-/-} and Paxillin-rescued *yap1*^{-/-};*yap1b*^{-/-} embryos. **(E)** Maximum projection of DAPI stained nuclei (lateral) in stage 16 WT, *yap1*^{-/-};*yap1b*^{-/-} and Paxillin-rescued *yap1*^{-/-};*yap1b*^{-/-} embryos. **(F)** Quantification of average nuclei flatness, which refers to the ratio between the second and the third axis of an ellipsoid, in WT, *yap1*^{-/-};*yap1b*^{-/-} and Paxillin-rescued *yap1*^{-/-};*yap1b*^{-/-} embryos. **(G)** Quantification of average nuclei sphericity in WT, *yap1*^{-/-};*yap1b*^{-/-} and Paxillin-rescued *yap1*^{-/-};*yap1b*^{-/-} embryos. Boxes represent quartiles; whiskers indicate maximum minimum values; points indicate independent embryos ($N^{\text{embryos}} = 4$). P-values are indicated in the figure. Two-sided Student's t-tests were performed to evaluate statistical significance. Scale bars = 5 μm .

This additional evidence has been now included in the revised version of the manuscript (please see Track changes for text changes). We are confident that this rescue experiment (FigS7), together with our previous findings (Fig 6) and the data available on the Yap mechanism of action in different cellular contexts (reviewed in Totaro et al Nat cell Biol 2018), provide enough support for the mechano-regulatory loop model we are proposing.

3. The authors drew their conclusion on YAP's control of mechanical properties based on their inference from the nuclear shape and the actin reporter Utrophin::GFP (Fig. 5, 6A, B). However, nuclear shape does not always correlate with cell compression, and actin over-polymerization leads to lower tissue tension (Pinto et al., Dev Cell, 2012). To make such a statement, they must carry out direct measurements, e.g. laser cutting of actin filaments and micro-pipette analysis.

R1: To support our claim of Yap proteins being involved in a mechano-regulatory loop we are providing mechanical stimulation experiments in the revised version of the work. These experiments, using a customized Univert mechanical tester for embryo compression show that Yap-dependent transcriptional program responds to mechanical tension (see new Figures 9 and S8). The converse regulatory arrow (i.e from Yap activation to intracellular tension) was inferred in our work from nuclear morphology, actin cortical recruitment, and focal adhesions assembly in mutant tissues; and it is now further supported by new measurements of total FA length, cell compactness index, and filopodial protrusions number (Figures 6, 7 and S6). Direct measurements showing a significantly reduced tissue tension have been already performed both in medaka *yap1* mutants and zebrafish gastrulating embryos, using micropipette aspiration and laser ablation experiments respectively (Porazinski. et al 2015). We are now mentioning these experiments that support our conclusions.

“From these measurements, we hypothesized that the noticeable reduction of cortical filamentous actin and FAs observed in *yap* mutant cells may lead to a decrease in intracellular tension, which is reflected in a more relaxed and rounded cell morphology. This is in agreement with previous observations indicating a reduced tissue tension in medaka mutants for *yap1* and *yap/taz* knockdown embryos in zebrafish (Porazinski. et al 2015)”.

The authors inferred mechanical properties via F-actin, FA accumulation and nuclear shape. However, this inference is misleading. For example, F-actin accumulation and its cell cortex tension do not correlate linearly. As F-actin accumulates, the cell tension increases then decreases (Chugh *et al.*, Nat Cell Biol 2015). Although nuclear deformation is required for YAP nuclear localization, deformed nuclei do not necessarily have nuclear YAP. Thus, nuclear deformation can not be used for inferring tissue mechanics. Thus, direct measurement of the mechanical properties of migrating cells is definitely required to support their conclusion and model. While this reviewer suggested direct measurement of the mechanical properties of cells in the previous review, but the authors did not deal with this suggestion. This reviewer stresses that they have to show alteration of

mechanical properties by direct measurements e.g. micro- pipet aspiration done in tissues (ref.19) and in single cells of zebrafish e.g. Kardash *et al.*, Nat Cell Biol 2010.

Although Porazinski *et al.*, demonstrated reduced TISSUE tension of the blastomere ring and the neural tube, they did not measure CELL tension in migrating cells during gastrulation. It is possible in fish embryos by micropipette aspiration as shown previously (e.g. Kardash *et al.*, Nat Cell Biol 2010).

R2: In our work we are inferring a reduced intracellular tension in mutant cells on the bases of reduced FAs, reduced F-actin accumulation, and reduced nuclear flatness. Mind we are not referring in the text to cortical tension: the article cited by the reviewer (we assume Chugh et al., Nat Cell Biol 2017, PMID: 28530659) analyzes surface tension dynamics during cell cycle progression, but not adhesion-dependent intracellular tension. Setting an adapted micropipette aspiration device would allow monitoring intracellular tension in the context of live migratory cells. Performing the experiments in detached/dissociated cells will not be informative, as maintaining the ECM-FA-actin cytoskeleton axis is essential to measure adhesion-dependent intracellular tension. However, we think that setting such a micropipette aspiration system for live monitoring is a very complex task, and certainly beyond the scope of this work.

A second readout of a reduced intracellular tension is the activation status of Yap by itself (Elosegui-Artola et al, Nat cell Biology 2016; Elosegui-Artola et al; Cell 2017). In our work, we are not basing our hypothesis exclusively in changes in nuclear shape, as we have also monitored Yap activation in wild type, compressed and mutant cells. On the light of all our observations, together with the reduced tissue tension previously reported in gastrulating fish embryos (ref 19), we are still convinced that the most parsimonious scenario is a reduced intracellular tension in *yap* mutant cells.

Minor points

1. In the hippo-YAP field, we use YAP rather than Yap e.g. in the review by Guan. (Ann Rev Biochem, 2019)

We thank the reviewer for bringing our attention to this issue: In this work we are using the zebrafish nomenclature suggested by the Genetic Nomenclature Guidelines (Trends in Genetics 1998; see also Zfin); which follows the system: zebrafish /*shha*/ Shha; human / *SHH* / SHH; mouse / *Shh* / SHH

We have scanned again the manuscript for potential inconsistencies and identified a few mistakes that were corrected (please see Track changes document).

2. P2. L15, P3. L9 “acto-myosin” should be actomyosin.

This has been corrected in the revised version (please see Track changes document).

3. P9 L29, Spell out WNT-PC should be spelt out.

This spelling mistake has been corrected in the revised version (please see Track changes document).

REVIEWERS' COMMENTS

Reviewer #1 (Remarks to the Author):

The second version of the revised manuscript "A Yap-dependent mechano-regulatory loop is essential during cell migration for embryo axis assembly" by Ana Sousa-Ortega, Javier Vazquez-Marin, Juan R. Martinez-Morales and colleagues proposed a response to each of the 3 points raised by this reviewer.

Point 1- The possibility of hypoxia and the question of how the controls were performed to reproduce the same degree of confinement on the uncompressed embryos as on the compressed embryos during direct mechanical stimulation experiments, received a convincing response. In addition, the missing information on the treatment of the controls was added in the Methods section.

Points 2 and 3 received a partial response only:

Point 2- Indeed, the demonstration of a mechanosensitive biological feature requires, when possible, both its reduction under defective mechanical conditions (here the application of different drugs), and its rescue by the application of mechanical stress (here direct compression) from the latter defective mechanical conditions (here at the migration stage). Using the Rock inhibitor, the experiment does not show such a result. This can be interpreted by the requirement of tension for the Yap pathway to be mechanosensitive (as the authors do), or by the fact that tension does not control the activation of the Yap pathway. The fact that different drugs have been used to inhibit tension favours the authors hypothesis but cannot substitute to a rescue experiment and prevents a definitive conclusion. This should have been discussed more extensively in the Discussion paragraph.

Point 3- In addition, the question of a role of Yap in maintaining tissue integrity (Porazinski et al Nature 2015) in response to morphogenetic forces, thereby permissively allowing a correct migration instead of actively feedbacking in the production of active forces involved in migration, remains a possibility. This should have been discussed as well in the Discussion paragraph.

The authors should discuss these points more broadly p11 of the manuscript after the citation of ref 19. The authors should also remove the word "loop" from the title and from their main conclusion. A title like "A Yap-dependent mechanoregulatory pathway is essential during ..." to let open the possibility to the Yap-mechanosensitive permissive model.

Reviewer #3 (Remarks to the Author):

Please see attached file.

Manuscript: NCOMMS-22-22583A: A Yap-dependent mechano-regulatory loop directs cell migration for embryo axis assembly: Ana Sousa-Ortega 1*, Javier Vázquez-Marín 1*, Estefanía Sanabria-Reinoso 1, Jorge Corbacho¹, Rocío Polvillo 1, Alejandro Campoy-López 1, Lorena Buono 1, Felix Loosli 2, María Almuedo-Castillo 1#, Juan R. Martínez-Morales 1#.

REVIEWER COMMENTS (Point by Point answer)

Reviewer #3 (Remarks to the Author):

Revision.1

In this report, Sousa-Ortega et al analysed yap1 and yap1b double mutant embryos that did not properly form the body axis during gastrulation. RNA sequencing, together with the DamID data they previously reported, showed that yap1 and yap1b are required for the expression of genes involved in cytoskeleton reorganization and cell-cell and cell-matrix adhesion. They next showed that Yap was active in cells migrating to the midline. In those cells, the amount of actin and the number of FAs were reduced, and their nuclei were more spherical. Finally, inhibition of Rho/Rock during gastrulation lead to rounder nuclei and a reduction in Yap-mediated transcription. They concluded that Yap activity regulates FAs and actomyosin, and that this mechanism forms a positive feedback loop.

This is a follow-up study of their previous report that provides some useful information. In their previous report, the authors identified Yap and Yap1b binding sites by using the DamID approach. They demonstrated that Yap and Yap1b have transcriptional targets controlling actin and FA formation and claimed that Yap controls mechanical properties of cells during gastrulation and forms a feedback loop. However, it has previously been shown that YAP acts as a mechano-effector from the analysis of the medaka YAP1 mutant by Porazinski et al (Porazinski et al, Nature 2015) and together with YAP's role as a mechano-transducer, they proposed the concept of a mechanical feedback loop. Furthermore, the author's claims about "direct target genes" and "feedback loop" are not fully supported by their results.

R1: We have carefully addressed all the comments raised. In the revised version, we are now providing mechanical stimulation experiments that strengthen our conclusions on Yap controlling a mechano-regulatory loop that directs cell migration during embryo axis assembly. In addition, we have polished our arguments on Yap-dependent transcriptional activation and correct all minor suggestions (Please see Track changes document).

Revision2:

The authors addressed the major points 1 and 4, but did not adequately address 2 and 3. They did not deal with this reviewer's concerns about their proposed mechanical model and mechanical property measurements. Therefore, the added data in this revision do not fully support the positive feedback "mechano-regulatory loop" model drawn by the authors.

R2: In the revised version we have made an effort to address all the remaining concerns, either by providing additional experiments or by clarifying our arguments in the text and the reply letter (please see also the Track changes document).

Revision3:

This reviewer does acknowledge the effort to address the points 2 and 3. However, these data do not fully support their model of the mechanical loop. Furthermore, despite of their response to this reviewer's points, they did not accordingly amend the texts. They have to amend the texts and the title according to the discussion between the authors and this reviewer.

Major points:

Major points 2. In the section titled “Yap senses and activates intracellular tension...”, the authors claim to show that Yap activity regulates FAs and the actomyosin cytoskeleton, which allows for the coupling of intracellular tension to the ECM and suggest the existence of mechano-regulatory feedback. Porazinski et al. previously reported the Yap’s newly discovered function as a mechano-effector in which medaka YAP mutants showed reduction of tissue tension and aberrant ECM formation. They also suggested the presence of a mechano-regulatory feedback together with Yap’s previously reported mechano-transducer functions. Furthermore, the author’s description that “Yap senses...”. is not correct. Yap is a mechano-transducer, but it is not a mechano-sensor (Dupont et al., Nature 2011).

R1: The work by Porazinski and coworkers describing the phenotype of *yap1* single mutants in medaka, is certainly a reference in the field as it highlighted the important role of Yap proteins in maintaining tissue morphogenesis and intracellular tension *in vivo*. We have referred to this article several times in the text. However, there are fundamental differences between the mechanical feedback loop suggested by Porazinski et al and that described in our work. Whereas Porazinski et al reported an *arhgap18*-dependent **negative** feedback loop involved in organogenesis (particularly during optic cup development), the Yap-dependent mechano-regulatory loop here described is a self-sustained positive loop active in migratory cells converging towards the midline. It is important to emphasize that our work about the role of Yap proteins focus on an earlier gastrulating stage that was not described previously in the work of Porazinski *et al.*, since in order to observe a clear phenotypic difference at this stage is required to analyze *yap1/yap1b* double mutants and the work of Porazinski et al. is limited to *yap1* single mutants.

Revision 2: The authors falsely claim that an earlier gastrulating stage phenotype was not described previously in the work of Porazinsky *et al.* Indeed, they, previously reported in Fig.1b and in quantitative analysis in c that the medaka YAP1 mutants exhibited **delayed blastopore closure and reduced F-actin in the envelope layer (EVL) during gastrulation.** They also showed that zebrafish YAP and TAZ double **knockdown embryos exhibited delayed blastopore closure** and even reduction of actomyosin ring contraction by mechanical measurement using laser cutting. Therefore, the author’s findings of the early gastrulating phenotype in YAP mutants are limited.

R2: The main defects observed in our double *yap1;yap1b* medaka mutants are the lack of directed cell migration and axis assembly, and these defects are not observed in single *yap1* medaka mutants. That is why we claim that we uncover new functions of Yap paralogs during medaka gastrulation. Despite the very valued descriptions of the Yap single mutant defects in Porazinski et al., we believe our findings are novel, since we described a new role shared by both *yap* paralogs in the embryo’s primary axis assembly, which is key for the construction of the vertebrate body plan. Moreover, we also described a mechanosensitive mechanism for the convergent migration of the precursor cells to the midline.

Revision 3: As this reviewer pointed out as above (highlighted), the following description has to be amended as follows. P3 L32; “In contrast, in *yap* single mutants, actin staining appeared more diluted at the delayed blastopore margin **as reported previously^{19, 23}**, and a decreased density of cells at the midline was observed”

Revision 2: In the revised version, the authors placed even more stress on the “**mechano-regulatory loop**” by changing the title to: “A Yap-dependent mechano-regulatory loop directs cell migration for embryo axis assembly”. In the abstract, “Our results **indicate** that Yap coordinates a **mechano-regulatory loop** to sustain intracellular tension and maintain the directed cell migration for embryo axis development”.

Unfortunately, these conclusions on the mechano-regulatory loop were not fully supported because of the lack of data showing that the YAP target gene mediates a mechano-regulatory “loop”. There are many other potential mechanisms by which YAP activation could lead to actomyosin contraction.

- (1) Firstly, the authors did not clarify what are the components of the “mechano-regulatory loop” in the Results section. This reviewer assumes that the loop could be ① mechanical stimuli > ② YAP activation > ③ *marcks1b* transcription > ④ (data linking ③ & ⑤ missing) > ⑤ F-actin accumulation > ⑥ actomyosin contraction (cell mechanical properties not measured) > ① as shown in this reviewer’s figure below. To clarify this, they have to describe components of the feedback loop in the Results section.

- (2) While the authors carried out the following 3 experiments, these experiments do not fully support their model of mechano-regulatory positive feedback because of the lack of evidence that YAP and actomyosin is linked by the target gene as shown ④ in this reviewer’s figure.

The 3 experiments were to examine whether mechanical positive feedback is operating in migrating cells, as shown in (B) this reviewer’s figure. Modulation of mechanical properties by (a) reduction of cortical tension by inhibiting actomyosin contraction using Rockout (blocking ⑥-①), (b) increased F-actin polymerization by CalyculinA (activating ⑥-①), and (c) mechanical compression (①), inactivated YAP in (a) or activated YAP in (b, c), and reduction or induction of ③ *marcks1b* expression.

To fill up the gap, the authors have to show that *marcks1b* overexpression rescues the YAP double mutant phenotype.

- (3) What is the molecular mechanism that links *marcks1b* and actin localization or actomyosin contraction? They cited the ref. 34, but the molecular mechanism by which *marcks1b* controls actomyosin contraction is not well understood. Thus, the authors have to show the mechanism,

e.g. as shown in ref.19 that ARHGAP18 suppresses F-actin polymerization to optimize actomyosin contraction.

- (4) P6 L23. While the authors showed that *marcks1b* expression coincided with YAP activation in lateral cells and its expression was reduced in YAP double mutants, it remains unclear whether *marcks1b* is indeed required for the cortical actin phenotype and cell migrations, and also for activating YAP if **positive feedback** works.

To distinguish (a) compression activate YAP via **positive** feedback mediated by *marcks1b* from (b) alterations of cell mechanical properties activate YAP and *marcks1b* as reported in many papers, the authors have to show that *marcks1b* overexpression could increase actomyosin contraction and YAP activation in WT embryos and conversely, knock-down of *marcks1b* leads to a reduction of cortical actin & cell tension, and YAP inactivation.

R2: The referee is depicting an oversimplified and lineal model of Yap mechanism of action. This model does not correspond to our view, as according to our data *marcks1b* is not the main/single effector of Yap. We follow the expression of this gene only as a readout of Yap activity, in addition to our Yap reporter 4xGTIIC::GFP. In fact, a key aspect of our work is precisely the description of an entire Yap-dependent transcriptional program that includes the parallel recruitment of hundreds of genes involved in ECM assembly, cytoskeletal organization, integrin binding, and focal adhesion (see Fig. 1 for the reviewers, Fig. 3, Table S1, and Supplementary dataset 1). This parallel recruitment draws a picture that is far from the lineal pathway outlined in the referee scheme.

Figure 1 for the reviewers: Main components of the Yap-dependent transcriptional program encode for proteins that provide a link between the ECM and the actin cytoskeleton.

In our view, it would be unlikely to expect a full rescue of the yap double mutant phenotype by *marcks1b* overexpression, or its phenocopy by *marcks1b* knockdown, as many other downstream regulators of the actin cytoskeleton act in parallel. Nevertheless, in order to explore this possibility, we decided to perform a loss-of-function approach, which is less prompt to out-of-context artifacts than overexpression experiments. We took advantage of the CRISPR/Cas13d that allows the specific and efficient targeting of mRNAs, and has been proved as an efficient tool in teleost models, including medaka (Kushawah et al 2020, Dev Cell). Knockdown of *marcks1b* using Cas13d together with specific gRNAs resulted in an efficient depletion of its mRNA. However, injected embryos displayed only a very mild phenotype, including delayed development and epiboly in 26% of the injected embryos. In all cases, the primary axis was correctly form in *marcks1b* depleted embryos (Fig2 for the reviewers). We conclude from this experiment that the Yap-dependent transcriptional program plays a broader role than merely the activation of *marcks1b* to direct axial formation.

Going back to the reviewer scheme, we strongly believe that the missing connection between Yap activation (2) and cortical actin accumulation (5) is simply the integrin-dependent assembly of focal adhesions (FAs). The link between integrin-mediated cell adhesion to the ECM and F-actin accumulation has been demonstrated extensively in the literature and can be considered textbook knowledge. see for example the classical work by Hynes and Destree, 1978 in Cell, and the reviews by Hynes Cell 1992, Schartz and Gardel 2012 J. Cell Science, Bachir et al. 2017 Cold Spring Harb Perspect Biol; Martino et al. Front in Physiol 2018.

In our work we showed that FAs, preferentially oriented towards the YSL in dorsally converging cells, are largely reduced in the double mutants (Fig 6). As the link between FAs and cortical actin accumulation is well-established, we reasoned that the reduced cortical F-actin, cell spreading and filopodia observed in the double mutants (Fig6 and S6) resulted from the deficient assembly of FAs. To prove this hypothesis, we have performed an additional experiment (now Fig S7, see attached) aimed to “rescue” the missing FAs in the double mutant embryos. As restoring all FAs components in the proper stoichiometric ratio is an experiment beyond our capacity, we decided to overexpress the FA component paxillin, a central node

in the ECM-FA-Actin cytoskeleton axis. Overexpression of paxillin::mKate mRNA at 300 pg per embryo (i.e. a higher concentration than in our previous analytical experiments), was now sufficient to partially rescue the mutant defects at the cellular level: including spread cell morphology, cell compactness, filopodia number and nuclear morphology (Fig S7).

Figure S7: Paxillin overexpression rescues cell and nuclear morphology in yap mutant embryos.

Revision 3:

1. Since the authors did not clarify their hypothesis on the YAP target genes which control cell migration, and the section of "YAP is active in migratory cells..." it reads that marcks1b could be the major YAP target (i.e. linear pathway) controlling cell migration. Therefore, the authors have to clarify their model by which YAP activates transcription of many FA related genes as yellow highlighted above at the end of the section of "YAP is active in migratory cells..."
2. This reviewer acknowledges author's effort, but the data altogether do not support their mechanical feedback model.
 - a. Fig. 1 for the reviewers does not include *paxillin* as the YAP target. If *paxillin* is not a YAP target gene, this experiment does not make sense.
 - b. Did paxillin overexpression rescued the cell migration phenotype, the most important indicator of axis formation?

3. Their feedback model in Fig. 9D is too simplified. If the authors still stick to the mechano-regulatory model, they have to add more detailed model as the reviewer's figure indicating each components including the green highlighted parts as above, together with the Figure1 for the reviewers in the supplement figures.
4. This reviewer is not yet convinced about their mechanical feedback model as shown in reviewer's figure 3.

Their results demonstrated that ① mechanical cues ② activates YAP which ③ controls src-F-actin-FA controlling genes to promote ⑦FA-ECM adhesion. However, this is simply one directional mechanical response.

To close a mechanical loop, the evidence by which ⑧ YAP activation leads to alterations in ECM mechanical cues is missing.

This is why this reviewer requested mechanical measurement of ECM. Although the authors cited the report of Porazinski *et al.* the authors claimed that their proposed mechanism is different from the mechanism of Porazinski *et al.*, which is a negative-feedback and also works at the different stages.

5. Please clarify what is the evidence for the positive feedback which cannot be explained by the simple mechano-sensory response.

Together, if the authors are not able to address the above point 2-5, they have to amend the title to "A YAP-dependent mechanical response directs cell migration for embryo axis assembly" and also amend the texts accordingly.

Point 3. The authors drew their conclusion on YAP's control of mechanical properties based on their inference from the nuclear shape and the actin reporter Utrophin::GFP (Fig. 5, 6A, B). However, nuclear shape does not always correlate with cell compression, and actin over-polymerization leads to lower tissue tension (Pinto *et al.*, Dev Cell, 2012). To make such a statement, they must carry out direct measurements, e.g. laser cutting of actin filaments and micro-pipette analysis.

R2: To support our claim of Yap proteins being involved in a mechano-regulatory loop we are providing mechanical stimulation experiments in the revised version of the work. These experiments, using a customized Univert mechanical tester for embryo compression show that Yap-dependent transcriptional

program responds to mechanical tension (see new Figures 9 and S8). The converse regulatory arrow (i.e from Yap activation to intracellular tension) was inferred in our work from nuclear morphology, actin cortical recruitment, and focal adhesions assembly in mutant tissues; and it is now further supported by new measurements of total FA length, cell compactness index, and filopodial protrusions number (Figures 6, 7 and S6). Direct measurements showing a significantly reduced tissue tension have been already performed both in medaka *yap1* mutants and zebrafish gastrulating embryos, using micropipette aspiration and laser ablation experiments respectively (Porazinski. et al 2015). We are now mentioning these experiments that support our conclusions.

“From these measurements, we hypothesized that the noticeable reduction of cortical filamentous actin and FAs observed in yap mutant cells may lead to a decrease in intracellular tension, which is reflected in a more relaxed and rounded cell morphology. This is in agreement with previous observations indicating a reduced tissue tension in medaka mutants for yap1 and yap/taz knockdown embryos in zebrafish (Porazinski. et al 2015)”.

Please see above section why this reviewer requested mechanical measurement of ECM if the authors claim the mechanical-loop.

Manuscript: NCOMMS-22-22583B: “A Yap-dependent mechanoregulatory program sustains cell migration for embryo axis assembly”: Ana Sousa-Ortega ^{1*}, Javier Vázquez-Marín ^{1*}, Estefanía Sanabria-Reinoso ¹, Jorge Corbacho ¹, Rocío Polvillo ¹, Alejandro Campoy-López ¹, Lorena Buono ¹, Felix Loosli ², María Almuedo-Castillo ^{1#}, Juan R. Martínez-Morales ^{1#}.

Point-by point reply letter

Reviewer #1 (Remarks to the Author):

The second version of the revised manuscript "A Yap-dependent mechano-regulatory loop is essential during cell migration for embryo axis assembly" by Ana Sousa-Ortega, Javier Vazquez-Marín, Juan R. Martínez-Morales and colleagues proposed a response to each of the 3 points raised by this reviewer.

Point 1- The possibility of hypoxia and the question of how the controls were performed to reproduce the same degree of confinement on the uncompressed embryos as on the compressed embryos during direct mechanical stimulation experiments, received a convincing response. In addition, the missing information on the treatment of the controls was added in the Methods section.

We are glad to read that the reviewer is now satisfied with our response.

Points 2 and 3 received a partial response only:

Point 2- Indeed, the demonstration of a mechanosensitive biological feature requires, when possible, both its reduction under defective mechanical conditions (here the application of different drugs), and its rescue by the application of mechanical stress (here direct compression) from the latter defective mechanical conditions (here at the migration stage). Using the Rock inhibitor, the experiment does not show such a result. This can be interpreted by the requirement of tension for the Yap pathway to be mechanosensitive (as the authors do), or by the fact that tension does not control the activation of the Yap pathway. The fact that different drugs have been used to inhibit tension favors the authors hypothesis but cannot substitute to a rescue experiment and prevents a definitive conclusion. This should have been discussed more extensively in the Discussion paragraph. The reviewer refers here to the experiment described in fig S9D, in which knockout treatment of the embryos largely prevents the compression-induced activation of Yap downstream targets. We already had toned down our claims in the previous version to address this point: “This observation is in agreement with our previous observations (Fig 8), and most probably indicates that Yap ability to mechanotransduce relies on the generation of intracellular tension”.

Furthermore, we disagree with the reviewer’s interpretation on the expected outcome of this experiment. Maintaining the functional acto-myosin cytoskeleton has been shown as an absolute requirement for Yap activation (Dupont 2016. Experimental Cell Research). Therefore, if mechanical information cannot be transmitted to the nucleus for yap activation upon knockout treatment, it is not logic to expect a full rescue of Yap mechanosensitive properties simply by applying exogenous tension. To make a mechanistic metaphor of knockout-inhibited embryos: the situation would be like a smooth ball rolling on a low-friction conveyor belt, in which increasing the speed of the belt would have little impact on the ball displacement. Of note, a very modest but significant “mechanical rescue” was observed in those experiments for the expression of *lamc1* and *marckl1b* expression.

Considering all evidence in the literature, together with our own data both using different drugs to inhibit/activate intracellular tension and performing embryo compression experiments, we think that the hypothesis of “tension not controlling the activation of the Yap pathway” is extremely unlikely.

Point 3- In addition, the question of a role of Yap in maintaining tissue integrity (Porazinski et al Nature 2015) in response to morphogenetic forces, thereby permissively allowing a correct migration instead of actively feedbacking in the production of active forces involved in migration, remains a possibility. This should have been discussed as well in the Discussion paragraph.

The authors should discuss these points more broadly p11 of the manuscript after the citation of ref 19. The authors should also remove the word "loop" from the title and from their main conclusion. A title like "A Yap-dependent mechanoregulatory pathway is essential during ..." to let open the possibility to the Yap-mechanosensitive permissive model.

These comments are in line with some suggestions by referee 3. In the revised version we have both removed the word loop from the title and elaborated on alternative hypothesis to the mechanoregulatory loop in the discussion section. The following sentences have been added after reference 19:

"An alternative to our proposed YAP-dependent mechanoregulatory loop in migratory cells is that Yap has a generic role in maintaining tissue integrity in the entire embryo¹⁹, and in consequence just a permissive function in cell migration. However, we observed that YAP is specifically active in migratory cells at gastrulation stages and shuts down at the embryo midline, where cells are densely packed and do not longer move extensively. Therefore, our results favor the scenario where Yap plays a unique function in migratory precursors due to their distinct geometrical and mechanical features."

Reviewer #3 (Remarks to the Author):

This reviewer does acknowledge the effort to address the points 2 and 3. However, these data do not fully support their model of the mechanical loop. Furthermore, despite of their response to this reviewer's points, they did not accordingly amend the texts. They have to amend the texts and the title according to the discussion between the authors and this reviewer.

We are glad that this reviewer acknowledges our efforts to address all questions raised. In the revised version we have both removed the word loop from the title and elaborated on alternative hypothesis to the mechanoregulatory loop in the discussion section. The following sentences have been added after reference 19. Nevertheless, we would like to emphasize that although alternative scenarios are now contemplated in the discussion, we still think that enough experimental evidence to support a mechanoregulatory loop as a favored hypothesis has been provided. Please see our comments below.

Point 2: As this reviewer pointed out as above (highlighted), the following description has to be amended as follows. P3 L32; "In contrast, in yap single mutants, actin staining appeared more diluted at the delayed blastopore margin as reported previously^{19, 23}, and a decreased density of cells at the midline was observed" We have modified the sentence in the revised version of the manuscript.

Point 3:

1. Since the authors did not clarify their hypothesis on the YAP target genes which control cell migration, and the section of "YAP is active in migratory cells..." it reads that marcks1b could be the major YAP target (i.e. linear pathway) controlling cell migration. Therefore, the authors have to clarify their model by which YAP activates transcription of many FA related genes as yellow highlighted above at the end of the section of "YAP is active in migratory cells..."

We partially disagree with the referee on this point. The section entitled "Yap is active in migratory cells" is preceded by another entitled "Yap transcriptional program primarily regulate cytoskeleton organization and cell adhesion components" in which the Yap-dependent program is described in detail. To emphasize this point we are now including the figure summarizing Yap downstream targets (Fig1 for the reviewers in the previous version) in the closing scheme in figure 9D.

2. This reviewer acknowledges author's effort, but the data altogether do not support their mechanical feedback model.

a. Fig. 1 for the reviewers does not include paxillin as the YAP target. If paxillin is not a YAP target gene, this experiment does not make sense.

b. Did paxillin overexpression rescued the cell migration phenotype, the most important indicator of axis formation?

The rationale behind using *paxillin mRNA* for the partial rescue experiment was as follows: First, we had experience with the overexpression of this *mRNA* and knew it was not toxic. More importantly, paxillin is a central component in focal adhesions, being the scaffold for many of the proteins that are regulated by Yap (e.g. talin, vinculin, actinin, zyxin or nexilin). Provided it was technically challenging to attempt a rescue experiment by simultaneous overexpression of *mRNAs* for several of these components, we decided to overexpress only *paxillin*, as a central hub in focal adhesions (Figure S7). We think the rationale of this experiment is valid.

When paxillin rescue experiments were performed, we focused our attention on the cellular aspects (i.e. cell and nuclear morphology. See Figure S7) that we could quantify with enough statistical power. We also observed a partial, though very clear, rescue of the axis formation in mutant embryos injected with Paxillin. The number of mutant embryos genotyped was not very high (n=3) and therefore we decided not to include these preliminary observations in the manuscript.

3. Their feedback model in Fig. 9D is too simplified.

If the authors still stick to the mechano-regulatory model, they have to add more detailed model as the reviewer's figure indicating each component including the green highlighted parts as above, together with the Figure 1 for the reviewers in the supplement figures.

We are now including this information in the final panel in Figure 9D

4. This reviewer is not yet convinced about their mechanical feedback model as shown in reviewer's figure 3. Their results demonstrated that ① mechanical cues ② activates YAP which ③ controls src-F-actin-FA controlling genes to promote ⑦ FA-ECM adhesion. However, this is simply one directional mechanical response.

To close a mechanical loop, the evidence by which ⑧ YAP activation leads to alterations in ECM mechanical cues is missing. This is why this reviewer requested mechanical measurement of ECM. Although the authors cited the report of Porazinski et al. the authors claimed that their proposed mechanism is different from the mechanism of Porazinski et al., which is a negative-feedback and also works at the different stages. Please clarify what is the evidence for the positive feedback which cannot be explained by the simple mechano-sensory response.

To answer this question it is important to take into account that our work provides a complete description of the Yap-activated program, showing the recruitment of a broad set of genes encoding for ECM and focal adhesion components (See summary in new Fig 9D). We strongly believe that the hypothesis of the ECM-FA mechanical coupling not being affected, upon the activation of all these genes, can be completely ruled out. Moreover, as we already mentioned, direct measurements showing a significantly reduced tissue tension have been already performed both in medaka *yap1* mutants and zebrafish gastrulating embryos, using micropipette aspiration and laser ablation experiments respectively (Porazinski. et al 2015).

We are still confident that on the light of all our findings, as well as data available on the Yap mechanism of action in different cellular contexts (reviewed in Totaro et al Nat cell Biol 2018), the mechanoregulatory loop model we are proposing is the most parsimonious scenario. Nevertheless, we have removed the term loop from the title and abstract and elaborated on alternative hypotheses in the discussion section.